# Toughening self-healing elastomer crosslinked by metal–ligand coordination through mixed counter anion dynamics

Hyunchang Park [1,2], Taewon Kang[1], Hyunjun Kim [1], Jeong-Chul Kim[3], Zhenan Bao [2] & Jiheong Kang [1] ✉

Mechanically tough and self-healable polymeric materials have found widespread applications in a sustainable future. However, coherent strategies for mechanically tough self-healing polymers are still lacking due to a trade-off relationship between mechanical robustness and viscoelasticity. Here, we disclose a toughening strategy for self-healing elastomers crosslinked by metal–ligand coordination. Emphasis was placed on the effects of counter anions on the dynamic mechanical behaviors of polymer networks. As the coordinating ability of the counter anion increases, the binding of the anion leads to slower dynamics, thus limiting the stretchability and increasing the stiffness. Additionally, multimodal anions that can have diverse coordination modes provide unexpected dynamicity. By simply mixing multimodal and non-coordinating anions, we found a significant synergistic effect on mechanical toughness (> 3 fold) and self-healing efficiency, which provides new insights into the design of coordination-based tough self-healing polymers.

Over recent decades, significant efforts have been devoted to developing intrinsically self-healing materials which can be readily repaired and used repeatedly upon mechanical damages for the realization of a sustainable future[1–6]. They are mostly based on dynamic covalent/non-covalent bonds, which can be repeatedly reformed after the reorganization of the functional groups at the break. To facilitate the reorganization process in a polymer matrix, amorphous and soft polymers having low glass transition temperature ($T_g$) and abundant dynamic bonds have been employed as polymer backbones and crosslinkers, respectively[7–9]. However, their dynamic nature derived from chain mobility and bond reversibility typically results in high viscoelasticity and low mechanical toughness. Recently, various approaches, including mixed strong/weak dynamic bonds[10,11], supramolecular polymerization of dynamic bonds[12,13], and robust but repairable dynamic bonds[14,15], have been reported to toughen the self-healing polymer. As a result, they have been used as new soft

material platforms for a broad range of emerging fields including soft robotics[16], biomedical devices[17,18], and energy storage devices[19,20].

Most examples of tough self-healing elastomers have been based on H (hydrogen)-bonding-based systems. Indeed, the combination of relatively weak hydrogen bonds, typically with binding energy ranging from 1 to 40 kJ/mol, allows the soft polymer network to have both toughness and dynamic behavior[21]. Nevertheless, it is challenging to understand the detailed mechanisms for their dynamic mechanical properties due to the complicated binding dynamics of weak H-bonding. Among the other dynamic bonds, metal–ligand coordination of which the typical bond strengths range from 100 to 200 kJ/mol is of particular interest[11,22–27]. Since it is more directional and predictable for bond formation compared to weaker H-bonding and its thermodynamic parameters possess rich tunability over a broad range, it would be a good model system to systematically understand the toughening mechanism of self-healing polymers[22,24]. Although a

[1]Department of Materials Science and Engineering, Korea Advanced Institute of Science and Technology (KAIST), Daejeon 34141, Republic of Korea.
[2]Department of Chemical Engineering, Stanford University, Stanford, CA 94305, USA. [3]Center for Nanomaterials and Chemical Reactions, Institute for Basic Science (IBS), Daejeon 34141, Republic of Korea. ✉e-mail: jiheongkang@kaist.ac.kr

number of previous investigations have contributed to the understanding of the metal−ligand crosslinked self-healing polymers, there is a lack of reports demonstrating significant mechanical toughening through fine-tuning the coordination environments (Fig. 1a).

Here, we present a design principle of tough self-healing polymer solely crosslinked by metal−ligand coordination. Key strategies are the deep understanding of counter-anion effects on dynamic mechanical properties and the mixing of two different counter anions to incorporate two discrete coordination dynamics. Since the principal role of counter anions is to maintain local charge neutrality, the effect thereof has often been less investigated. Even though several previous works have pointed out the counter anion effect[26,28,29], the focus was made simply on the coordination event of the counter anion, whether it coordinates or not. We utilize three classes of counter anions (Fig. 1b): (i) non-coordinating anion,

(ii) coordinating anion, and (iii) multimodal anion. A multimodal anion represents a coordinating anion that is capable of multiple coordination modes[30-32]. Each class exhibits characteristic mechanical properties derived from the effect of the counter anion on the coordination dynamics. Detailed studies on mechanical and dynamic properties elucidate that metal−ligand bond exchange, a dominant energy dissipation mechanism, is slower as the coordinating ability of the counter anion increases. The slower dynamics leads to higher modulus and strength but lower stretchability. Notably, multimodal anions have additional coordination modes that provide unanticipatedly high stretchability despite the strong coordination of the anion. More intriguingly, the synergy of non-coordinating and multimodal anions simultaneously enhances the mechanical toughness and self-healing efficiency despite a trade-off relationship between mechanical robustness and viscoelasticity.

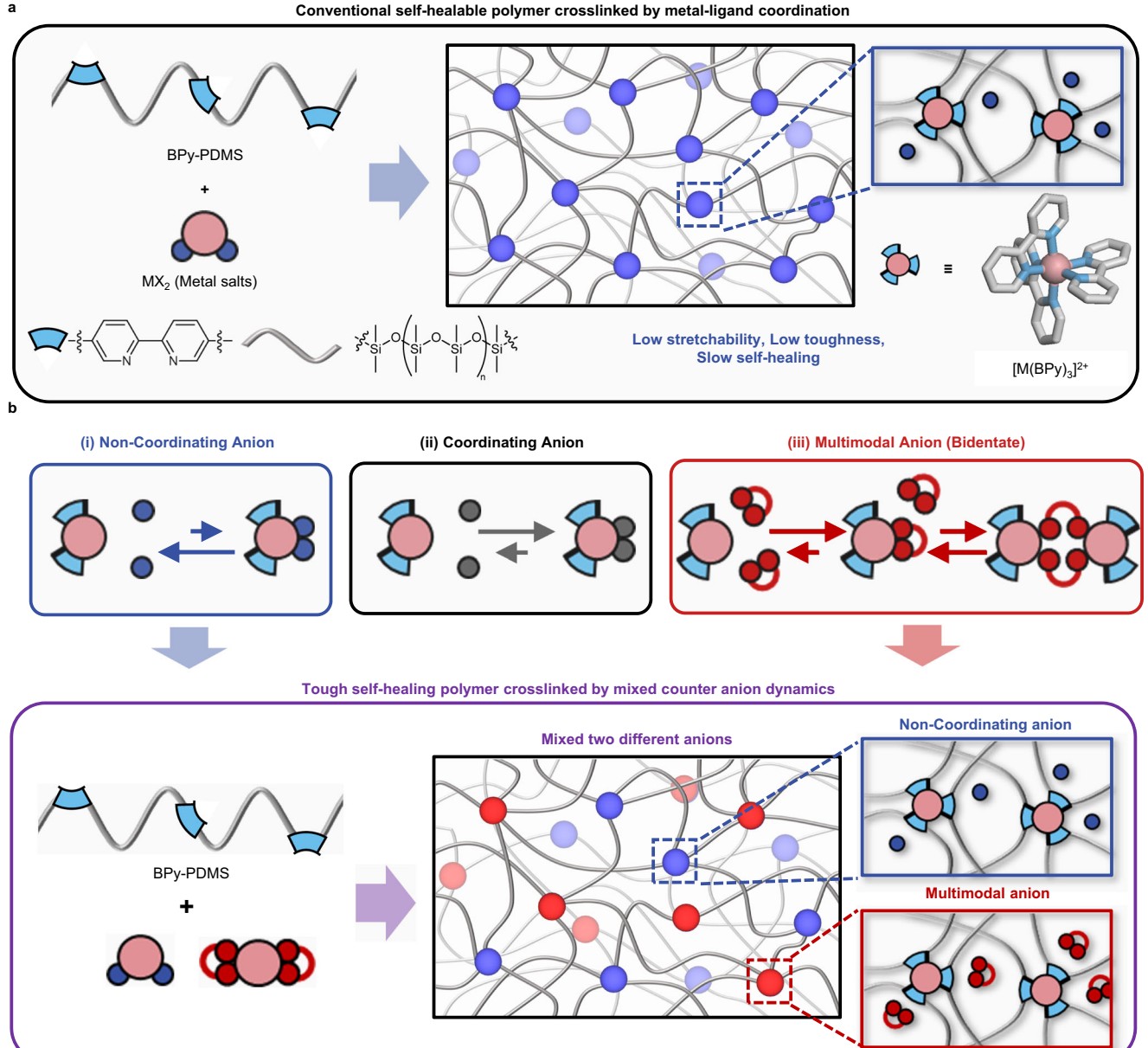

**Fig. 1 | Design strategy for toughening self-healing polymer solely crosslinked by metal−ligand coordination. a** Conventional self-healing polymer network crosslinked by coordination bonds. **b** Impact of different types of anions on the coordination environment (top) and toughening of self-healing polymer through mixing of non-coordinating and multimodal anions (bottom).

## Results

### Design of self-healing polymer and effect of counter anions on mechanical properties

As a model system, 2,2′-bipyridyl (BPy)-containing poly-dimethylsiloxane (PDMS) polymer (BPy-PDMS) was prepared by nucleophilic acyl substitution of 2,2′-bipyridine-5,5′-dicarbonyl dichloride with bis(3-aminopropyl) terminated PDMS (MW ~ 5000) according to previous reports (Supplementary Fig. 1)[26,33]. Bidentate BPy ligand can bind tightly to the metal center with a rigid 5-membered chelate ring. In addition, it is a charge-neutral ligand, thus making it a suitable ligand for examining the effect of counter anions. By treating BPy-PDMS with a stoichiometric amount (BPy:$Zn^{2+}$ = 3:1) of zinc salts delivered as $ZnX_2$ (X = counter anion), the polymers crosslinked by $Zn^{2+}$–BPy coordination were readily formed (Zn-X-BPy-PDMS). The formation of $Zn^{2+}$–BPy complexes was confirmed by solution UV-vis titration studies (Supplementary Figs. 2 and 3). The crosslinking by zinc(II) affords sufficient dynamicity to the system since it is a kinetically labile $d^{10}$ transition metal ion. Zinc(II) ion prefers a coordination number of six with octahedral geometry, yet can also be stabilized with a coordination number of four, which provides geometric flexibility as well[34]. For counter anions, we used conventional ones for each class: (i) non-coordinating anions: trifluoromethanesulfonate (OTf⁻) and bis(trifluoromethanesulfonyl)imide (TFSI⁻), (ii) coordinating anion: chloride (Cl⁻), (iii) multimodal anions: monodentate acetate (OAc⁻) and bidentate acetylacetonate (acac⁻). According to the spectrochemical series and the chelate effect[35,36], the order of coordination ability of the anions is as follows; acac⁻ > OAc⁻ > Cl⁻ > OTf⁻ > TFSI⁻.

Figure 2 depicts our hypothesized mechanism of anion-driven coordination bond dynamics in a self-healing polymeric matrix. When an external force is applied, the metal (M)–BPy coordination bonds of the crosslinkers are broken, making vacant coordination sites. M–BPy bond breakage and reformation process at these vacant sites, that is, the process of energy dissipation, would be greatly affected by the coordination dynamics of the anions, which manifests as a heavy anion dependence of mechanical properties (*vide infra*).

Thanks to effective crosslinking, all the polymers with various counter anions can be obtained as freestanding polymer films. Tensile stress–strain curves of the freestanding Zn-X-BPy-PDMS polymers at a loading rate of 100% min⁻¹ are shown in Fig. 3a. Although the only difference is the counter anion species, tensile strength (TS), Young's modulus (E), and fracture strain (ε) are significantly different between the films (Supplementary Table 1). Non-coordinating anions (OTf⁻, TFSI⁻) make the polymer network softer and more stretchable, whereas anions with coordinating ability (Cl⁻, OAc⁻, and acac⁻) give higher modulus and fracture strength. In addition, the residual strain after cyclic mechanical test with a strain of 50% was smaller as the coordination ability of the anion is greater (Fig. 3c and d), suggesting that the polymer networks become more elastic with a strong binding anion. Rheological behavior is consistent with the above observations. As shown in Fig. 3e, master curves obtained by frequency sweeps and time–temperature superposition (TTS) over a temperature range of 0 to 100 °C ($T_0$ = 20 °C) reveal that the crossover point (where tan($\delta$) = 1) is shifted to a lower frequency region as the anion binds to the metal ion more strongly. This means that the crosslinking network becomes more elastic and robust as the coordinating ability of the anion increases. From the slope of the plot obtained by fitting the horizontal shift factor, $a_T$, to the Arrhenius equation, the activation energy of flow transition was estimated (Supplementary Fig. 4). The value of flow activation energy was significantly higher for anions with coordinating ability (Cl⁻, OAc⁻, and acac⁻), indicating that it is harder to make the polymer network flow when the anion binds more tightly.

To elucidate the aforementioned anion-dependent mechanical properties, shear stress relaxation experiments were performed at a shear strain of 3% (Fig. 3b). The applied stress was relaxed rapidly for the case of non-coordinating anions (OTf⁻ and TFSI⁻), indicating fast

and efficient energy dissipation. In stark contrast, when the anions (Cl⁻, OAc⁻, and acac⁻) are able to coordinate with the metal ion, the value of stress relaxes gently due to slow energy dissipation. The capability of energy dissipation relies on the bond-breaking-reformation dynamics of M–BPy coordination[37]. Non-coordinating anions are unable to bind to vacant coordination sites (Fig. 2a). The vacant sites would thus be occupied by other BPy ligand detached from the metal center in close proximity. Such fast-dissociative ligand exchange, i.e., fast bond exchange, provides efficient energy dissipation, leading to high stretchability. When comparing Zn-OTf- and Zn-TFSI-BPy-PDMS, it is reasonable to attribute the lower tensile strength and higher stretchability of Zn-TFSI-BPy-PDMS to the size of the anion (Supplementary Table 2). The bulkiness of TFSI⁻ likely leads to a notable plasticizing effect when incorporated into the polymer network[38]. On the other hand, anions with coordinating ability can make bonds with the metal center at the vacant sites (Fig. 2b). Although the coordinating ability of the bidentate BPy ligand is superior to the coordinating anion, higher mobility of the small counter anions compared to the BPy ligands embedded in the polymer backbone enables its coordination with the metal. In addition, the counter anions would have a higher collision frequency with the metal ions due to its higher local concentration which is achieved by coulombic interactions. As a result, the metal center would be stabilized, which makes the M–BPy bond reformation slower than in the case of non-coordinating anions. Therefore, this M–anion binding impedes the relaxation of applied stress, thus leading to enhanced elasticity of the polymer films[39].

### Effect of diverse coordination modes

If we consider the coordination ability of counter anions as the only factor that affects the coordination dynamics, the stretchability of Zn-acac- and Zn-OAc-BPy PDMS should be lower compared to that of Zn-Cl-BPy-PDMS due to the less dynamic metal–ligand binding equilibrium. As shown in Fig. 3a, however, Zn-acac- and Zn-OAc-BPy-PDMS exhibit both better stretchability ( > 100%) and higher tensile strength than Zn-Cl-BPy-PDMS. We postulated that these intriguing mechanical properties might originate from the additional coordination modes of multimodal anions which can diversify the coordination environment (Fig. 2c). The hydrophobic nature of the PDMS backbone might exclude the charged metal centers, which in turn make them located close enough to have diverse coordination modes[23,40,41]. Oxygen-based ligands have been widely studied as having a variety of coordination modes due to their rich lone-pair electrons. One prominent example is the paddle-wheel structure of carboxylate ligands[42,43]. Within this context, the diverse coordination of acac⁻ and OAc⁻ are most likely to exist when two or more metal centers are in close contact.

To validate this suggestion, an analysis of relaxation time ($\tau$) and relaxation time spectrum (H($\tau$)) was carefully investigated. Viscoelastic materials can be described as n spring/dashpot elements connected in parallel[24,44,45]. Here, the relaxation time spectrum (H($\tau$)) is the strength of each spring element at a specific relaxation time, which is proportional to the degree of energy dissipation at the given relaxation time ($\tau$)[46]. Therefore, we can obtain information about the energy dissipation modes by estimating H($\tau$) from the experimentally measured values. From the experimental G′ and G″ values of the master curves of Zn-X-BPy-PDMS films, we conducted mathematical non-linear fitting to obtain best-fit lines of the master curves and corresponding relaxation time spectra[47,48]. From that, the contributions of energy dissipation modes were calculated to evaluate the diversity in the energy dissipation mechanism (Supplementary Fig. 5). The number of dominant energy dissipation modes was fixed to three since the contribution of the fourth dissipation mode is negligible for all the polymers. The relaxation time spectra of Zn-acac-BPy-PDMS (Fig. 3f) and Zn-OAc-BPy-PDMS (Supplementary Fig. 6) display significantly broad and multiple peaks with the comparable contribution of the third component

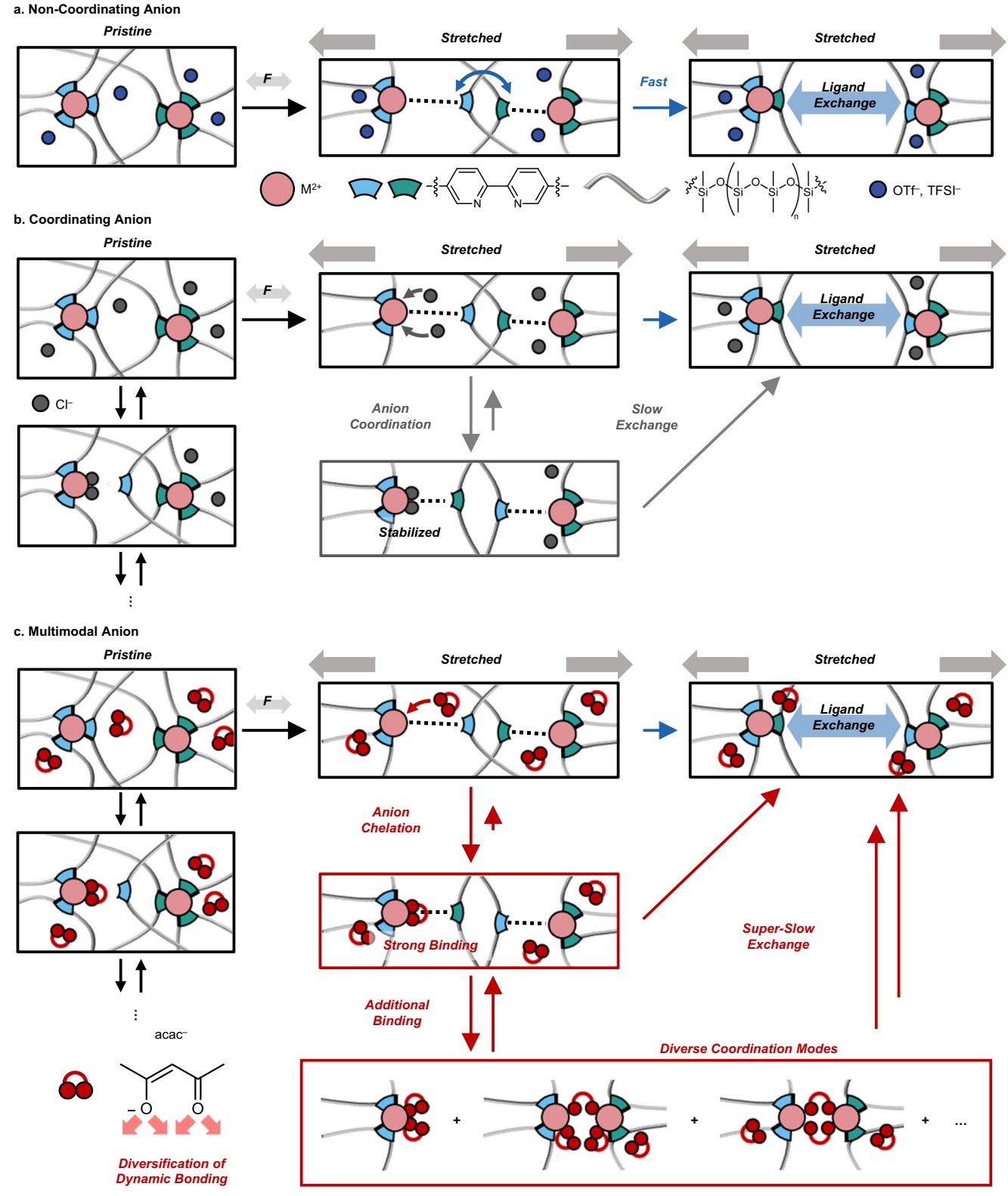

**Fig. 2 | Schematic illustration of hypothesized counter anion effects on mechanical properties and energy dissipation mechanisms.** Plausible energy dissipation mechanisms of Zn-X-BPy-PDMS polymers under mechanical stimuli;

**a** Non-coordinating anions (X = OTf⁻, TFSI⁻), **b** coordinating anion (X = Cl⁻), and **c** multimodal anions (X = OAc⁻, acac⁻).

(Supplementary Fig. 5), whereas the other three polymers exhibit a nearly single peak with a negligible third component (Fig. 3g, h, and Supplementary Figs. 5 and 6). This result clearly depicts the function of the multimodal coordination to diversify the energy-dissipating coordination modes.

In addition, small-angle X-ray (SAXS) and neutron (SANS) scattering data from the multimodal cases indicate broadening of the scattering peak at $q$ ~ 0.1 Å⁻¹ which corresponds to the ligand−ligand domain spacing (Supplementary Fig. 7)[28]. For ¹³C CP (Cross-polarization) / MAS (Magic angle spinning) solid-state NMR spectra of the

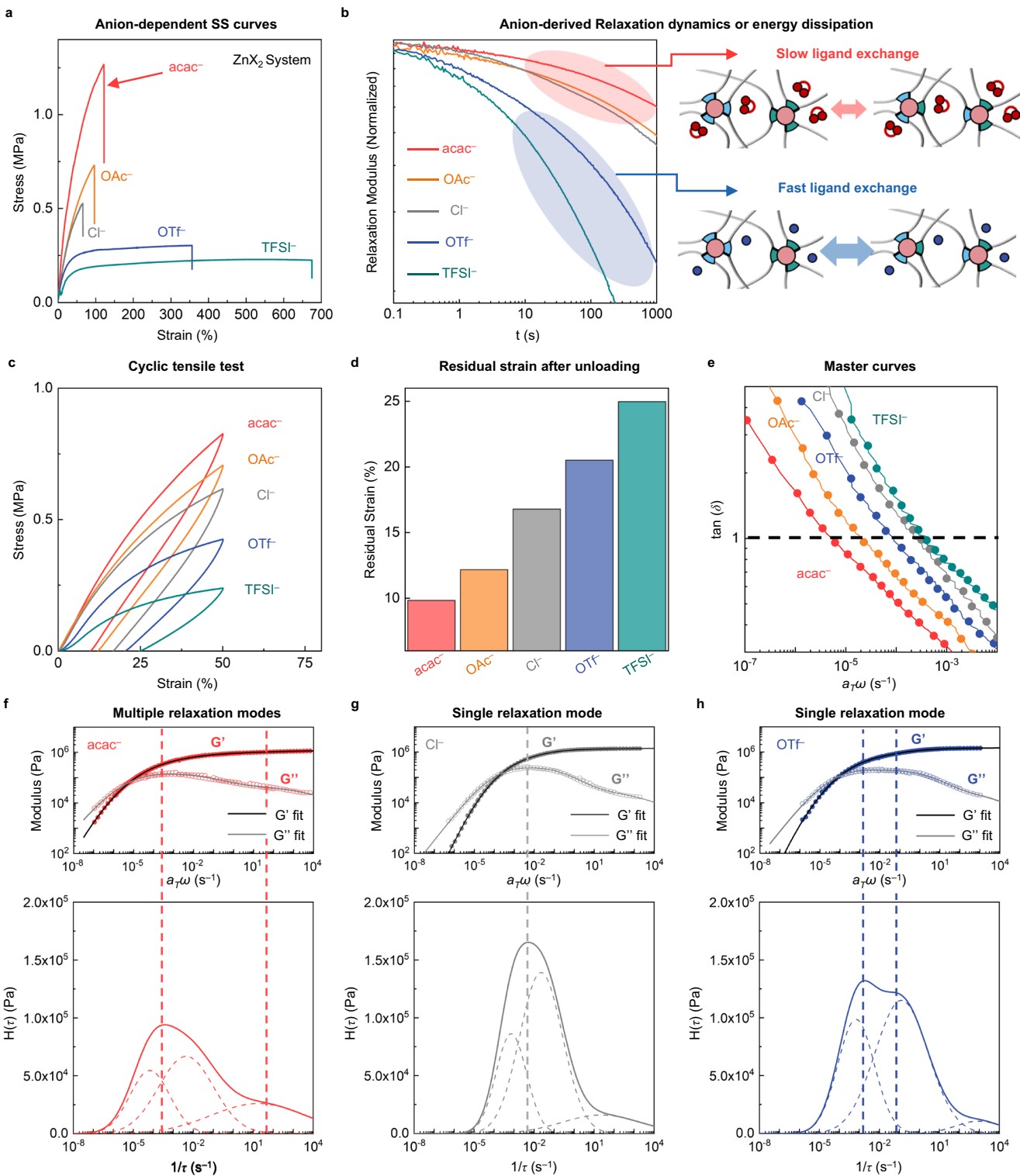

**Fig. 3 | Effects of counter anions on mechanical and dynamic properties of Zn-X-BPy-PDMS. a** Stress–strain curves of Zn-X-BPy-PDMS polymer films (acac⁻; red, OAc⁻; orange, Cl⁻; gray, OTf⁻; blue, TFSI⁻; green) with a sample width of 3 mm, a thickness of 0.2–0.4 mm, and a length of 5 mm at a loading rate of 5 mm min⁻¹. **b** Shear stress relaxation spectra of the polymer films under a shear strain of 3%. Coordinating anions show slower relaxation compared to non-coordinating anions. **c** Stress–strain curves of the polymer films with a sample width of 3 mm, a thickness of 0.2–0.4 mm, and a length of 5 mm under cyclic loading (loading rate:

10 mm min⁻¹). **d** Residual strains after unloading. **e** Master curves showing frequency-dependent tan(δ) obtained by time–temperature superposition (TTS). **f**–**h**, top, Experimental master curves (dots) and best-fit lines (G′; black line, G″; gray line) of **f** Zn-acac-BPy-PDMS (red), **g** Zn-Cl-BPy-PDMS (gray), and **h** Zn-OTf-BPy-PDMS (blue). **f**–**h**, bottom, Relaxation time spectra, H(τ) of **f** Zn-acac-BPy-PDMS, **g** Zn-Cl-BPy-PDMS, and **h** Zn-OTf-BPy-PDMS. The dashed lines in the plot indicate the spectra of each energy dissipation mode.

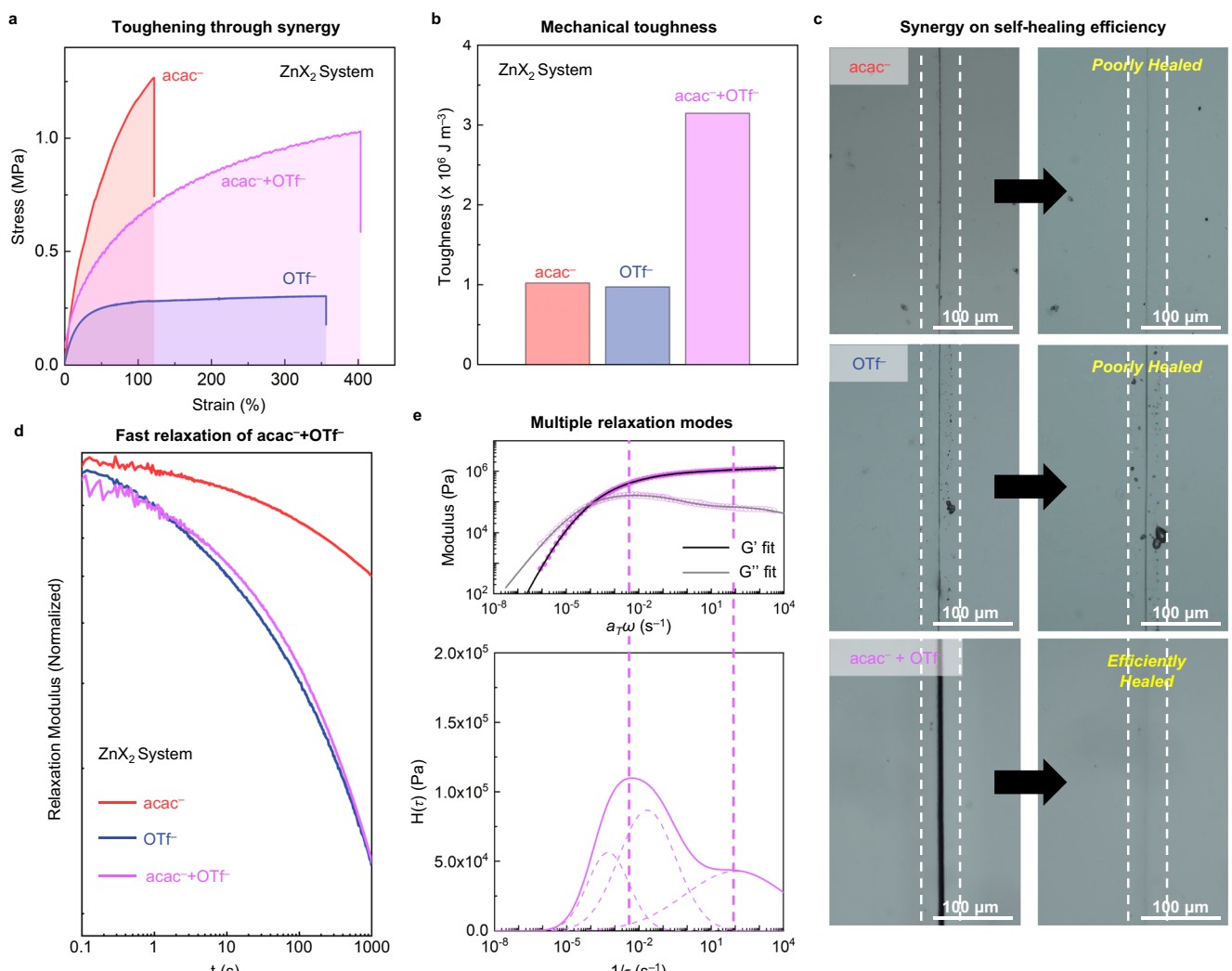

**Fig. 4 | Synergistic effect on mechanical and dynamic properties of non-coordinating (OTf⁻) and multimodal (acac⁻) anions. a** Stress–strain curves of polymer films comprising acac⁻ (red), OTf⁻ (blue), and a 1:1 mixture of acac⁻ and OTf⁻ (pink) (a sample width of 3 mm, a thickness of 0.2–0.4 mm, and a length of 5 mm at a loading rate of 5 mm min⁻¹). **b** Tensile toughness showing a synergistic effect of mixing acac⁻ and OTf⁻. **c** Optical microscope images of damaged (left) and healed (right) films. The disappearance of the scar was observed with the case of mixed anion after 6 h at 50 °C, while the scar remained with the cases of single anions under identical conditions. **d** Shear stress relaxation spectra of polymer films under a shear strain of 3%. The relaxation spectrum of Zn-acac-OTf-BPy-PDMS is almost superimposable with the spectrum of Zn-OTf-BPy-PDMS. **e**, top, Experimental master curve (pink dots) and best-fit lines (G'; black line, G''; gray line) of Zn-acac-OTf-BPy-PDMS. **e**, bottom, Relaxation time spectrum, H(τ) of Zn-acac-OTf-BPy-PDMS which exhibits acac⁻-like broad and multiple peaks. The dashed lines in the plot indicate the spectra of each energy dissipation mode.

polymer films, significant peak broadening of aromatic carbon resonances was also observed with multimodal anions (Supplementary Fig. 8). Such peak broadening might be a result of diverse coordination environments. Film UV-vis absorption provides additional experimental evidence for the diverse coordination. The absorption spectra of those multimodal systems show a band at $\lambda = 400$ nm region, which is responsible for the light-yellow color of Zn-acac- and Zn-OAc-BPy-PDMS films. In contrast, the other three don't show any absorption at the visible region (Supplementary Fig. 9). To delineate the origin of the longer wavelength absorption, we performed DFT (B3LYP/6-31 + + G(d,p)) geometry optimization of acac-bound zinc(II) model complexes. As shown in Supplementary Figs. 10 and 11, HOMOs of the acac-bound complexes are localized at acac⁻, and LUMOs are localized at BPy ligands. It can be predicted that the longest wavelength absorption of Zn-acac-BPy-PDMS corresponds to the ligand-to-ligand charge transfer (LLCT) band, thus implying the participation of multimodal anions in the coordination sphere.

## Synergistic effect of non-coordinating and multimodal anions
To toughen a material, it is required to make improvements simultaneously in seemingly conflicting functions, i.e., mechanical strength, and stretchability. We accomplished significant toughening of our Zn–BPy crosslinked polymer by simply mixing non-coordinating OTf⁻ and multimodal acac⁻ anions. The polymer film of mixed acac⁻ and OTf⁻ anions (Zn-acac-OTf-BPy-PDMS) was prepared by crosslinking in a molar ratio of BPy:Zn(acac)₂:Zn(OTf)₂ = 6:1:1. Synthetically, it is straightforward when a comparison is made with all organic polymer systems which require complicated and troublesome synthetic modification.

As illustrated in Fig. 4, we conducted comparative studies on the mechanical and dynamic properties of Zn-acac-OTf-BPy-PDMS film with single anion films. Intriguingly, the tensile strength of Zn-acac-OTf-BPy-PDMS (1.03 MPa) was still as high as that of Zn-acac-BPy-PDMS even though the amount of acac⁻ is reduced by half (Fig. 4a). Moreover, it stretched more (403%) than Zn-OTf-BPy-PDMS, thus resulting in significantly higher mechanical toughness (3.15 × 10⁶ J m⁻³) than the sum of Zn-acac- and Zn-OTf-BPy-PDMS (Fig. 4b).

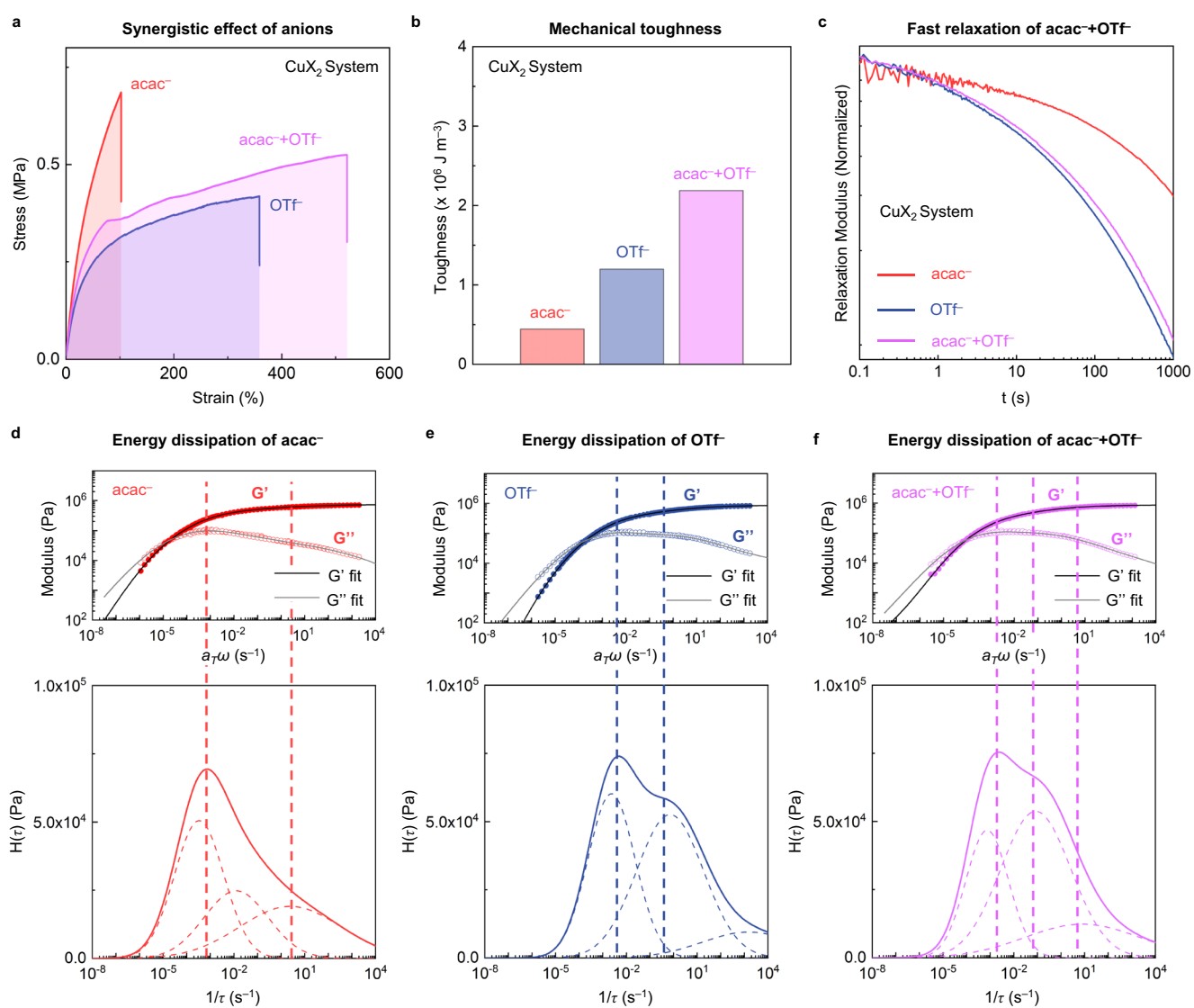

**Fig. 5 | Effects of counter anions on mechanical and dynamic properties of Cu-X-BPy-PDMS. a** Stress–Strain curves of Cu-X-BPy-PDMS polymer films comprising acac⁻ (red), OTf⁻ (blue), and a 1:1 mixture of acac⁻ and OTf⁻ (pink) (a sample width of 3 mm, a thickness of 0.3–0.4 mm, and a length of 5 mm at a loading rate of 5 mm min⁻¹). **b** Tensile toughness showing a synergistic effect of mixing acac⁻ and OTf⁻. **c** Shear stress relaxation spectra of polymer films under a shear strain of 3%. The relaxation spectrum of Cu-acac-OTf-BPy-PDMS is almost superimposable with the spectrum of Cu-OTf-BPy-PDMS. **d–f**, top, Experimental master curves (dots) and best-fit lines (G'; black line, G"; gray line) of **d** Cu-acac-BPy-PDMS (red), **e** Cu-OTf-BPy-PDMS (blue), and **f** Cu-acac-OTf-BPy-PDMS (pink). **d–f**, bottom, Relaxation time spectra, H(τ) of **d** Cu-acac-BPy-PDMS, **e** Cu-OTf-BPy-PDMS, and **f** Cu-acac-OTf-BPy-PDMS. The dashed lines in the plot indicate the spectra of each energy dissipation mode.

This synergy also impacts the self-healing of the damaged film. Optical microscope images showed that the Zn-acac-OTf-BPy-PDMS film damaged by a razor blade underwent complete healing after being heated at 50 °C for 6 h, while the other films were not successful in healing under identical conditions (Fig. 4c). To quantitatively compare the self-healing efficiency, we measured temperature-dependent healing behavior by using a rheometer according to the previously reported method (see Methods for details on the procedure)[13,49]. As shown in Supplementary Fig. 12, the Zn-acac-OTf-BPy-PDMS film starts to heal at a lower temperature (< 20 °C) and more effectively recovers its intrinsic tensile strength (78% at 70 °C) compared to the single anion systems. This denotes the synergy of acac⁻ and OTf⁻ markedly increases the self-healing efficiency.

Toward a deeper understanding of the synergistic effect, the stress relaxation profile of Zn-acac-OTf-BPy-PDMS was examined. Surprisingly, it was almost superimposable with the spectrum of Zn-OTf-BPy-PDMS, even though the amount of OTf⁻ anion is half that of

Zn-OTf-BPy-PDMS (Fig. 4d). Furthermore, SAXS and SANS scattering patterns were also superimposable with that of Zn-OTf-BPy-PDMS (Supplementary Fig. 13). These results imply that OTf⁻ anion dominates the domain construction and energy dissipation, thereby providing high stretchability as Zn-OTf-BPy-PDMS. On the other hand, the relaxation time spectrum obtained from the master curve of Zn-acac-OTf-BPy-PDMS shows broad and multiple peaks similar to that of Zn-acac-BPy-PDMS (Fig. 3e and Supplementary Fig. 14), which suggests the diverse coordination environment derived from acac⁻ anion. The contribution of acac⁻ was also detected by the development of an absorption band around 400 nm with the increasing equivalent of acac⁻ (Supplementary Fig. 15). These two functions, energy dissipation and coordination diversification, work concurrently and synergistically, providing stretchability of Zn-acac-OTf-BPy-PDMS even higher than that of Zn-OTf-BPy-PDMS. We concluded that the improvement in self-healing efficiency also originates from this synergy in coordination dynamics.

In contrast, no synergistic effect of acac⁻ and TFSI⁻ on mechanical properties was observed (Supplementary Fig. 16). The shear stress relaxation spectrum and the SAXS profile of Zn-acac-TFSI-BPy-PDMS are intermediate between those of Zn-acac-BPy-PDMS and Zn-TFSI-BPy-PDMS. That is, the energy dissipation of Zn-acac-TFSI-BPy-PDMS is less effective than the case of Zn-TFSI-BPy-PDMS, which is responsible for its poor stretchability. Although the relaxation time spectrum of Zn-acac-TFSI-BPy-PDMS is similar to that of Zn-acac-BPy-PDMS, the function of the non-coordinating TFSI⁻ anion in this mixed anion system is not sufficient to elicit a positive synergistic effect. The absence of synergy is presumably due to the steric bulkiness of TFSI⁻ anion (Supplementary Table 2), which makes the anions difficult to coexist successfully within the polymer matrix. Mixing the coordinating (Cl⁻) and noncoordinating (OTf⁻ or TFSI⁻) anions does not produce a synergistic effect on the mechanical properties as well (Supplementary Fig. 17). It is most likely because the coordinating and non-coordinating anions act in opposite directions and interfere with each other; the coordinating anions disturb the ligand-exchange process, while the non-coordinating anions do not. Accordingly, to be synergistic, it is required that the anions with clearly distinct functions operate independently and do not interfere with each other.

### Counter anion effects on the Cu²⁺-crosslinked system

To confirm the applicability of anion-dependent coordination dynamics to the system with other kinetically labile transition metal ions, we prepared the polymers crosslinked by Cu²⁺–BPy coordination bonds (Cu-X-BPy-PDMS) and investigated whether there are differences in mechanical properties depending on the counter anion or not (Fig. 5a and b). As in the case of zinc(II), Cu-X-BPy-PDMS polymers exhibit the same trend in anion dependence and synergistic behavior. From the shear stress relaxation spectra, we confirmed that Cu-OTf-BPy-PDMS dissipates the applied energy more effectively than Cu-acac-BPy-PDMS, and the mixed anion system Cu-acac-OTf-BPy-PDMS follows the relaxation feature of Cu-OTf-BPy-PDMS (Fig. 5c). The rheological tendency seems inconsistent with the zinc(II)-crosslinked system (Fig. 5d–f), however, the calculated contributions of each dissipation component of Cu-acac-BPy-PDMS show comparable values while Cu-OTf-BPy-PDMS shows the significantly low contribution of the third component (Supplementary Fig. 18). Also, we observed the increase in the third component contribution when OTf⁻ is mixed with acac⁻, indicating the synergistic effect of the counter anions.

## Discussion

This work demonstrates (i) significant effects of the counter anion on the mechanical and dynamic properties of the polymers crosslinked by metal complexation, and (ii) a strategy of mixing two different counter anions to toughen self-healing polymer. We found that the non-coordinating anions provide efficient energy dissipation through the rapid ligand-exchange process while the coordinating anions slow down the ligand-exchange by disturbing the reformation of metal–ligand bonds. Besides working as coordinating anions, multimodal anions function to diversify the coordination environment, which provides additional energy-releasing pathways. With a suitable combination of anions, we demonstrated the significant synergistic effect of non-coordinating and multimodal anions on mechanical toughening and self-healing efficiency, and figured out that such synergy exists regardless of the type of metal ion. We anticipate that the counter-anion effect will serve as a straightforward tool for fine-tuning and toughening coordination-based self-healing materials.

## Methods

### Mechanical characterization

All mechanical tensile experiments were conducted on an Instron 68SC-1 instrument. At least three samples were tested for each polymer film. Tensile tests were performed under ambient conditions with rectangular specimens of freestanding polymer films with approximate dimensions of 5 mm × 3 mm × 0.3 mm at a loading rate of 5 mm min⁻¹. Young's modulus was determined from the initial slope of the stress–strain curves. Cyclic tensile tests were conducted with samples of the same dimension at a loading rate of 10 mm min⁻¹ with a maximum strain of 50%. The cyclic measurements were iterated three times (Supplementary Fig. 19).

### Rheological characterization

All rheometric measurements were conducted on an Anton Paar MCR 302 rheometer with an 8 mm-diameter parallel plate setup and a Peltier temperature control system. Samples were cut into a disc shape with a diameter of 8–9 mm. Frequency sweeps for plotting master curves were performed with an applied shear strain of 1% and a normal force of 0.25 N in the angular frequency range of 0.1–100 rad/s at designated temperatures. The temperature was varied from 0 °C to 100 °C in 20 °C increments with a 180 s latency when the measuring temperature was reached. Time–temperature superposition (TTS) was carried out with RheoCompass software. For shear stress relaxation experiments, the shear stress over time was recorded under a fixed shear strain of 3% and a normal force of 0.25 N. For each measurement, a pre-strain step was performed for 5 mins with a shear strain of 0.1% to increase reproducibility.

### Method for calculating the relaxation time spectrum H(τ)

Relaxation time spectra were calculated from the experimentally obtained G′ and G″ values of the master curves. According to the regularization method, the relation between $H(\tau)$ and G′ or G″ can be expressed as integral equations: Eqs. 1 and 2 for G′ and G″, respectively, where $\omega = \tau^{-1}$.

$$G'(\omega) = \int_{-\infty}^{\infty} H(\tau) \frac{\omega^2 \tau^2}{1 + \omega^2 \tau^2} d\ln(\tau) \tag{1}$$

$$G''(\omega) = \int_{-\infty}^{\infty} H(\tau) \frac{\omega \tau}{1 + \omega^2 \tau^2} d\ln(\tau) \tag{2}$$

For the ease of mathematical fitting, an assumption was made that the spectrum of $H(\tau)$ of each energy dissipation component has a log-normal distribution with respect to $\tau$. Using MATLAB software, a nonlinear iterative optimization algorithm was implemented by the *lsqcurvefit* function. For the log-normal distribution equation of $H(\tau)$ with $n$ terms (Eq. 3), proper guesses based on the experimental results were first made to the initial value, lower and upper bounds for a total of $3n$ coefficients $A_i$, $\tau_i$, and $\sigma_i$.

$$H(\tau) = A_1 \exp\left(-\frac{(\ln(\tau) - \ln(\tau_1))^2}{2\sigma_1^2}\right) + A_2 \exp\left(-\frac{(\ln(\tau) - \ln(\tau_2))^2}{2\sigma_2^2}\right)$$
$$+ A_3 \exp\left(-\frac{(\ln(\tau) - \ln(\tau_3))^2}{2\sigma_3^2}\right) + \dots \tag{3}$$

With initial values, the algorithm operated to minimize the logarithmically weighted sum of the squares of the differences between the calculated and experimental data, as in below equation:

$$\min_{(coeffs)} \sum_{\omega_i} \{\log_{10}(G'_{fit}(\omega_i)) - \log_{10}(G'_{data}(\omega_i))\}^2$$
$$+ \{\log_{10}(G''_{fit}(\omega_i)) - \log_{10}(G''_{data}(\omega_i))\}^2 \tag{4}$$

where $G_{fit}$ is the storage and loss moduli calculated from Eqs. (1) to (4), and $G_{data}$ is the experimentally obtained values. This non-linear fitting process affords the fitted master curves and corresponding $H(\tau)$. The contribution of each dissipation component was calculated by

integrating the distribution curve and normalizing it so that the sum of the contributions of all components is 1.

## Method for self-healing tests

Temperature-dependent healing tests were conducted on a TA instrument ARES-G2 rheometer. A disk-shaped polymer film (diameter; 8 mm, thickness; ~0.3 mm) was glued on the stage, and another film of the identical size and shape was attached to the shaft. A square-shaped Teflon sheet (10 mm × 10 mm × 0.05 mm) having a hole (diameter; 1.5 mm) at the center was sandwiched between two films with a normal force of 0.05 N. The whole system was heated up to 130 °C for 30 min to merge the films at the hole in the Teflon sheet. The sample was cooled to r.t. and stabilized at r.t. for 5 min. Then, the merged films were separated by lifting the shaft at a rate of 1 mm s$^{-1}$. The separated films were attached again with a force of 0.05 N, heated to the designated temperature, and compressed with a normal force of 0.1 N for 1 h. After returning to r.t., the shaft was lifted up at a rate of 0.1 mm s$^{-1}$ while recording the force over time.

## Solid-state NMR characterization

Solid-state $^{13}C$ NMR spectra were measured at room temperature using a magic angle spinning (MAS) rate of 10 kHz with a Bruker Avance 400WB spectrometer operating at 400.13 MHz for $^1H$ and 100.61 MHz for $^{13}C$. $^{13}C$ CP/MAS NMR spectra were observed at a CP contact time of 4 ms, a pulse width of 2.5 μs (CP pulse sequence), and a relaxation delay of 5 s.

## Density Functional Theory (DFT) calculations

All density functional theory (DFT) calculations were conducted using Gaussian 16 program suite[50]. All geometry optimizations were carried out with B3LYP density functional and 6-31 + + G(d,p) basis set.

# Data availability

The authors declare that all data supporting the findings of this study are available within the article and its Supplementary Information files or from the corresponding author upon request.

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

## Acknowledgements

This study was supported by the National R&D Program through the National Research Foundation of Korea (NRF) funded by the Ministry of Science and ICT (Grant No. NRF-2021M3H4A1A04092882), and the Wearable Platform Materials Technology Center (WMC, Grant No. NRF-2022R1A5A6000846). This study was partially supported by the Industrial Strategic Technology Development Program-Alchemist Project (1415180859, Chiral perovskite LED smart contact lens based hyper vision metaverse) funded by the Ministry of Trade, Industry & Energy (MOTIE, Korea) and Korea Evaluation Institute of Industrial Technology (KEIT, Korea).

## Author contributions

H.P. and J.K. conceived and designed the experiments. H.P. synthesized and characterized metal–ligand-based self-healing elastomers with varying counter anions. T.K. performed mathematical fitting of the master curves for relaxation time spectrum analysis. H.P., T.K., and H.K. measured anion-dependent self-healing efficiency. J.-C.K. obtained $^{13}C$ and $^{1}H$ solid NMR spectra. H.P. and J.K. wrote the paper. Z.B. provided feedbacks on the paper and data analysis. All the authors discussed the results and commented on the manuscript. J.K. supervised the study.

## Competing interests

The authors declare no competing interest.
