## [Peer Review File · Nature Communications]

Toughening self-healing elastomer crosslinked by metal–ligand coordination through mixed counter anion dynamicsREVIEWER COMMENTS

Reviewer #1 (Remarks to the Author):

This manuscript aims to propose a new strategy to address the trade-off relationship between mechanical robustness and viscoelasticity, based on the effects of counter anions on the coordination dynamics. By mixing bridging and non-coordinating anions, they achieved a significant synergistic effect on improving both the mechanical toughness and self-healing efficiency. Such results are interesting to a certain extent. However, several critical issues should be addressed before considering the publication of this manuscript in Nature Communications.

1) Some of the results seem to be self-contradictory. As the molar ratio of Zn²⁺ metal to BPy ligand is 1:3, tris(BPy)zinc(II) coordination complexes are expected to be dominant in the pristine polymer matrix, since BPy ligand has stronger binding ability to metal center as compared to the counter anions investigated in this study (maybe acetylacetonate is an exception). Then, why Zn-acac-BPy-PDMS and Zn-OTf-BPy-PDMS films show the ligand-to-ligand charge transfer (LLCT) band in the UV spectra (LLCT occurs only when both ligands are coordinated to the same metal ion)? If OAc⁻ and acac⁻ coordinate to the metal center before stretching, then the crosslinking density of Zn-acac-BPy-PDMS and Zn-OTf-BPy-PDMS would be lower (because OAc⁻ and acac⁻ cannot function as crosslinking units), why the mechanical strength of Zn-acac-BPy-PDMS and Zn-OTf-BPy-PDMS is higher than Zn-Cl-BPy-PDMS? In a previous study, the Young's modulus of polymers with different counter anions are comparable (J. Am. Chem. Soc. 2016, 138, 6020–6027).

2) The differences in mechanical and self-healing properties of polymers with different counter anions are all attributed to the different coordination dynamics. However, other factors may also contribute to these differences. For example, it is known that bulky counter anions will result in polymer chains with more free volumes and enhanced mobility, which also lead to lower mechanical strength but higher stretchability (ACS Appl. Mater. Interfaces 2018, 10, 2105–2113). It is also possible that the differences of mechanical and self-healing properties in these polymers are not correlated to the coordination dynamics unless the author can provide more convincing evidence.

3) The author claimed that “the bridging coordination of acac⁻ and OAc⁻ are most likely to exist when two or more metal centers are in close contact.” It is true that acetate (OAc⁻) and acetylacetonate (acac⁻) can bind to two metal ions in bridging mode, but that happens only when steric hindrance is not a problem (no bulky ligands are coordinated to the metal center). In this study, if pyridinyl ligands (especially when two bpy ligands) have coordinated to the metal center, I don’t think acetate (OAc⁻) and acetylacetonate (acac⁻) can exist as bridging ligands.

4) Page 3, line 46-48, “Indeed, H-bonding motifs have a wider range of bonding modes and strength compared to other dynamic bonds such as metal–ligand coordination, π – π interaction, and dynamic covalent bond.” Actually, it is generally believed that metal–ligand coordination has wider range of bonding modes and strength compared to H-bonding.

5) The performance of polymers reported in this manuscript is not very competitive. In spite of their relative mechanical strength (about 1 MPa), they only self-heal at elevated temperature or for quite long time. Many better self-healing polymers have been reported previously.

Reviewer #2 (Remarks to the Author):

In their manuscript entitled "Toughening self-healing elastomer crosslinked by metal-ligand coordination through mixed counter anion dynamics", Park et al. present a intriguing study of the use of what they call bridging and non-coordinating ions to influence the mechanical properties of elastomers. Although the concept is interesting, and potentially adds new value to the growing literature on such materials, there are a number of items that should be addressed before publication

1. Authors refer several times to the impact of their work on a "sustainable society" in the abstract and introduction. This is unclear and should be better explained or removed.

2. Lines 48-51. It is not obvious how the binding dynamics of H-bonding are more complicated than those of metal-ion complexes as described in this work. Please clarify.

3. Line 55. There have been many studies of metal-ion coordinated binding in polymer networks, including studies of ion type, mixed ions, and of the chemical environment, notably by Holten-Anderson, among others. A google scholar search of metal ion coordination in elastomers returns over 20,000 results. Authors should be much clearer about how their results provide substantially new information, and should be careful of claims of novelty given the large literature.

4. Lines 70-75. Claims here are vague. Define "extra dynamicity", "synergistic actions" and what is meant by "universal toughening" given the limited material set tested here.

5. Line 83. What is the evidence of tris coordination? Is there a distinct spectroscopic signal that you can provide?

6. In the mechanical analysis, starting on ~Line 99, it is unclear how the authors selected the strain rates for tensile testing, and the extent to which the report parameters may depend on strain. Loading rates of 100% per minute are reported, which appear to be extremely fast, and may engage the fast relaxation time scales reported later. Do what extent are the elastic modulus, tensile strength and fracture strain dependent on the strain rate? Can the

differences in anion type be attributed to differences in strain rate relative to the intrinsic relaxation time scales of the system? Can authors compare the expected differences in energetic bonding given the anions selected?

7. Line 127. Define stretchability and relate this to the terms presented earlier (in particular, fracture strain). Explain more clearly how you relate this to toughness and dissipation.

8. Lines 127-129. Authors suggest that kinetic competition between diffusion and a coordination reaction may play a role in network dynamics -- this should be expanded and clarified.

9. Lines 197-200. Authors should provide more information about the optical methods used, as well as the rheometry measurements. It is difficult to understand the argument as written.

10. Line 215. The evidence for the role diversification of the coordination environment is not unclear. Is this conjecture based on observations, or is there specific evidence of difference here? Authors should be clearer about delineating results and interpretation.

11. Claims of universality are not supported since only one polymer system has been tested. This should be rephrased.

12. Were the complexes and materials sensitive to water or the presence of other solvents?

13. Figure 1: In panel B it is not clear what authors mean by "dynamics" and how they judge fast versus slow. In particular, panels (B i) and (B ii) are identical except for the arrow direction -- how does directionality factor into descriptions of dynamics? Also, why is (B iii) considered super-slow? And, how are any of these considered tris coordinated? In the lowest panel, right hand side, how are any of the bridging anions playing a role (red box)?

14. Figure 2: In panel B, why are the anion coordination and slow exchange transitions irreversible? (same question for panel C). Can Raman spectroscopy or imaging determine

any of these transitions or complexes more quantitatively?

15. Figure 3: In panel A, how repeatable are these curves? How many replicates were performed? In panel C, what was the effect of loading rate, and waiting time before reloading? Did you see a change in residual strain when loading dynamics were altered?

Reviewer #3 (Remarks to the Author):

This manuscript brings insightful design strategy for self healing polymers using metal-ligand coordination. Overall the manuscript is in the scope of Nature Communications and can be considered for publication after addressing or clarification of the comments. My specific comments are below:

1) line 112: It's a bit unclear why shift of crossover point to a lower frequency region means that coordinating anions render the crosslinking network more elastic and robust. Does it mean that the crosslinking network is more elastic when measuring properties at the same rate ?

2) In terms of wording, it seems like bridging anions are grouped with coordinating anions creating some confusion at some instances for example in line 112 and 115. Should it be "bridging anions and coordinating anions" instead of "coordinating anions" in line 112 and 115 ?

3) line 138: Authors mention that Zn-acac- and Zn-OAc-BPy-PDMS exhibit better stretchability than Zn-Cl-BPy-PDMS. It is clear as the highest strain in Fig 3a is larger for the former. However, it is difficult to see why Zn-Cl-BPy-PDMS is said to have higher tensile strength than others. Could the authors please clarify as the peak stress for Zn-Cl-BPy-PDMS is slightly higher than Zn-OAc-BPy-PDMS?

4) In Fig 3f and Fig 5d, it is a bit unclear how the second peak is identified as the second peak is not visually apparent. Could the authors clarify the existence of second peak?

5) typo: line 433 'on a TA instrument'

Point-to-Point Responses to Reviewers' Comments

Reviewer #1

This manuscript aims to propose a new strategy to address the trade-off relationship between mechanical robustness and viscoelasticity, based on the effects of counter anions on the coordination dynamics. By mixing bridging and non-coordinating anions, they achieved a significant synergistic effect on improving both the mechanical toughness and self-healing efficiency. Such results are interesting to a certain extent. However, several critical issues should be addressed before considering the publication of this manuscript in Nature Communications.

=> We thank the reviewer for careful reading of our manuscript. The reviewer commented that “*Such results are interesting to a certain extent*”, but pointed out that there are “*several critical issues should be addressed*”.

=> We have carefully revised the manuscript, and our response to the reviewer's helpful comments/suggestions is provided below. All main revisions are marked in red font.

1) Some of the results seem to be self-contradictory. As the molar ratio of Zn^{2+} metal to BPy ligand is 1:3, tris(BPy)zinc(II) coordination complexes is expected to be dominant in the pristine polymer matrix, since BPy ligand has stronger binding ability to metal center as compared to the counter anions investigated in this study (maybe acetylacetonate is an exception). Then, why Zn-acac-BPy-PDMS and Zn-OTf(OAc)-BPy-PDMS films show the ligand-to-ligand charge transfer (LLCT) band in the UV spectra (LLCT occurs only when both ligands are coordinated to the same metal ion)?

=> This is a legitimate point. The spectroscopic results including film UV-vis spectra and ^1H solution NMR spectra clearly indicate the participation of coordinating counter anions (acac^- and OAc^-) to the coordination sphere when the film is not stretched. The film UV-vis spectra exhibit small but clear absorption band at the visible region which corresponds to the LLCT (from counter anion to BPy ligand). In our hypothesized mechanism, however, we assumed that tris(BPy)zinc(II) coordination complexes are dominantly formed to 1) simplify the complex mechanism of the system, and 2) focus on the counter anion dynamics when strain is applied. We agree with the reviewer's comment. To address this issue, we replace **Fig. 2** with **the figure below** showing the equilibrium between tris(BPy)zinc(II) and anion-bound forms for the coordinating anions (**Fig. 2b and 2c**). In addition, we have revised the term "tris(BPy)zinc(II) coordination complexes" on line 83 of the main text to " Zn^{2+} -BPy coordination", and "tris(BPy)zinc(II) crosslinkers" on line 95 of the main text to "**the crosslinkers**".

a. Non-Coordinating Anion

b. Coordinating Anion

c. Multimodal Anion

Fig. 2 | Schematic illustration of hypothesized counter anion effects on mechanical properties and energy dissipation mechanisms. Plausible energy dissipation mechanisms of

Zn-X-BPy-PDMS polymers under mechanical stimuli; **a**, Non-coordinating anions ($X = \text{OTf}^-$, TFSI^-), **b**, coordinating anions ($X = \text{Cl}^-$), and **c**, multimodal anions ($X = \text{OAc}^-$, acac^-).

=> Although the coordinating counter anions (acac^- , OAc^- , and Cl^-) participate in the coordination sphere, the film UV-vis spectra (**Supplementary Fig. 10**) indicate that tris(BPy)zinc(II) coordination complexes are dominant. The major absorption peaks (200 ~ 350 nm) are similar to those of non-coordinating anion (OTf^- and TFSI^-), and the absorbance of the LLCT band (near 400 nm) is less than 10% of the absorbance of the highest peak (acac^- ; 10%, OAc^- ; 7%, Cl^- ; 10%). In addition, the solid state ^{13}C CP/MAS NMR spectra look almost identical except for the peak broadness for all the counter anions (**Supplementary Fig. 8**). Furthermore, the overall scattering profiles obtained from SAXS, WAXS and SANS experiments (**Fig. r1**) are almost superimposable for all the anions, indicating that the crosslinked polymer networks are similar at 0% strain regardless of the counter anion.

Fig. r1 | Overall a, SAXS, WAXS and b, SANS scattering patterns of Zn-X-BPy-PDMS films.

=> To obtain direct experimental evidence, we performed X-ray absorption spectroscopy (XAS) of the Zn^{2+} -BPy crosslinked polymer films, which are expected to reveal information about the local environment around the metal center (**Fig. r2**). We could find small differences according to the counter anions in XANES (X-ray Absorption Near Edge Structure) and EXAFS (Extended X-ray Absorption Fine Structure), however, there was a stability issue. The crosslinked films turned black within a minute of X-ray exposure. This X-ray beam damage makes it hard to ensure that the differences in the spectra come from the structural differences around the crosslinker.

Fig. r2 | XANES and EXAFS spectra of Zn-acac-, Zn-Cl-, and Zn-OTf-BPy-PDMS films.

=> We also performed solution UV-vis studies to further support our proposed mechanism (**Supplementary Figs. 2 and 3**). Due to the solubility difference of BPy-PDMS polymer and zinc salts, it was unable to try Job's analysis. Therefore, we conducted titration of a CHCl_3 solution of BPy-PDMS ($33 \mu\text{M}$) by continuously increasing the molar ratio of Zn^{2+} salt solutions (4.95 mM , solvent: EtOH for $\text{Zn}(\text{acac})_2$ and MeOH for the others). Regardless of the counter anion, no significant difference was observed up to the addition of 0.33 equivalent zinc salt (**Supplementary Fig. 2a, c, e, and Supplementary Fig. 3a, c**). We could observe well-defined isosbestic points for all the cases, which means the predominant formation of $\text{tris}(\text{BPy})\text{Zn}(\text{II})$ complex. The counter anion dependence is observed when the additions are above 0.33 equivalent. As the molar ratio of Zn^{2+} to BPy ligand exceeds 1:3, the isosbestic points for all the cases move to shorter wavelength, indicating the formation of Zn^{2+} complexes with different binding stoichiometry. For the non-coordinating counter anions (OTf^- and TFSI^-), the spectral changes above 0.33 eq are almost identical (**Supplementary Fig. 3b and d**). On the other hand, the spectra of coordinating anions (acac^- , OAc^- , and Cl^-) above 0.33 eq exhibit an apparent counter anion dependence (**Supplementary Fig. 2b, d, and f**), presumably

by the coordination of anions. For the case of acac^- , the emergence of absorption band under 300 nm with increasing amount of Zn^{2+} salt (**Supplementary Fig. 2a and b**) might be due to the absorption of free $\text{Zn}(\text{acac})_2$ salt (**Fig. r3**). It is true that the highly diluted solution state cannot represent the situation in the condensed phase. Combining the results from both film and solution UV-vis studies, however, we believe that the dominant coordination complex under the 1:3 ratio of Zn^{2+} to BPy is the tris(BPy)zinc(II) complex regardless of the binding ability of the counter anion.

Supplementary Fig. 2. UV-vis absorption spectra obtained by titration of a CHCl_3 solution of BPy-PDMS (33 μM) with Zn^{2+} salt solutions of coordinating counter anions. **a, b**, $\text{Zn}(\text{acac})_2$, **c, d**, $\text{Zn}(\text{OAc})_2$, **e, f**, ZnCl_2 .

Supplementary Fig. 3. UV-vis absorption spectra obtained by titration of a CHCl₃ solution of BPy-PDMS (33 μM) with Zn²⁺ salt solutions of non-coordinating counter anions. **a, b,** Zn(OTf)₂, **c, d,** Zn(TFSI)₂.

Fig. r3 | Normalized absorption spectrum of Zn(acac)₂ salt (33 μM) in CHCl₃/EtOH mixed

solvent (150:1 v/v).

Main text, Page 5

The predominant formation of Zn²⁺-BPy complexes was confirmed by solution UV-vis titration studies (Supplementary Figs. 2 and 3).

If OAc⁻ and acac⁻ coordinate to the metal center before stretching, then the crosslinking density of Zn-acac-BPy-PDMS and Zn-OTf-BPy-PDMS would be lower (because OAc⁻ and acac⁻ can not function as crosslinking units), why the mechanical strength of Zn-acac-BPy-PDMS and Zn-OTf(OAc)-BPy-PDMS is higher than Zn-Cl-BPy-PDMS?

=> Although the coordinating counter anions can bind to the metal center, the predominant species is the Zn-BPy coordination complex according to the experimental data that we mentioned above. In addition, the polymer network structure that is directly related to the crosslinking density is almost similar regardless of the counter anion (Fig. r1).

=> Assuming a situation in which the counter anion binds, we agree with the reviewer's point. The participation of the counter anion in the coordination sphere would reduce the crosslinking density. However, we found the major factor for the anion-dependent mechanical properties in the dynamics of metal-ligand reversible crosslinks under the applied stress based on the rheological stress relaxation studies. According to the literature (Parada, G. A., Zhao, X. Ideal reversible polymer networks. *Soft. Matter.* **14**, 5186–5196 (2018)) which provides theoretical explanation of the reversible polymer networks, the storage and loss modulus can be expressed as equations r1 and r2 when the reversible polymer network follows a Maxwell model of spring/dashpot elements.

$$G' = \nu_e K_B T \frac{\omega^2}{\omega^2 + (k_-)^2} \quad (\mathbf{r1})$$

$$G'' = \nu_e K_B T \frac{\omega k_-}{\omega^2 + (k_-)^2} \quad (\mathbf{r2})$$

Here, v_e is the concentration of elastically-active chains that has the same meaning as the ‘crosslinking density’ in the reviewer’s comment, and k_- is the relaxation rate constant. If we only consider the effect of crosslinking density (v_e), then it is true that the somewhat reduced crosslinking density in the case of coordinating anions results in low G' value and low elasticity. However, we also need to consider the effect of the relaxation dynamics (k_-). Experimentally, we confirmed that the better the coordination ability of the counter anion, the slower the stress relaxation rate (**Fig. 3b**). We proposed that this slower relaxation is derived from the stabilization of the dissociated species by coordination of the anion. The slower relaxation means lower k_- value and corresponding higher G' value according to the **equation r1**, which indicates that the polymer network is more elastic with the coordinating anion. This higher elasticity with stronger anion binding is the reason for the higher mechanical strength of Zn-acac-BPy-PDMS compared to the other polymer films even though the crosslinking density is lower. The order of mechanical strength ($\text{acac}^- > \text{OAc}^- \sim \text{Cl}^- > \text{OTf}^- > \text{TFSI}^-$) follows the order of coordinating ability of the counter anions.

In a previous study, the Young’s modulus of polymers with different counter anions are comparable (J. Am. Chem. Soc. 2016, 138, 6020–6027).

=> The Young’s modulus only considers the very first linear region of the stress–strain curve because viscoelastic polymers usually have very short linear elastic region under tensile stress. Accordingly, the comparison of the Young’s modulus value is often not that precise especially in the same order of magnitude. Therefore, we didn’t put emphasis on the exact values of the Young’s modulus. Instead of the Young’s modulus, we tried to explain the system with other experimental approaches including cyclic tensile test and detailed rheological studies.

=> Although we used the chemically same polymer as the previous study, the synthesized polymer could give different mechanical properties because it is afforded by step-growth

polymerization using macromonomers. The molecular weight, polydispersity index, crosslinking condition, and film preparation method of our system are different from the literature that the reviewer mentioned, resulting in a slight mismatch of the mechanical properties. However, the trend in the counter anion effect observed in this study aligns with the findings of the previous study. Additionally, the values of Young's modulus obtained in this study are within the same order of magnitude as those reported in the previous study.

2) The differences in mechanical and self-healing properties of polymers with different counter anions are all attributed to the different coordination dynamics. However, other factors may also contribute to these differences.

=> We thank the reviewer's valid point and totally agree that there might be other contributing factors in this highly complex condensed system. While it is challenging to attribute the entire system behavior to a single factor, we have identified the coordination dynamics as the primary factor that can account for the experimental results consistently.

For example, it is known that bulky counter anions will result in polymer chains with more free volumes and enhanced mobility, which also lead to lower mechanical strength but higher stretchability (ACS Appl. Mater. Interfaces 2018, 10, 2105–2113). It is also possible that the differences of mechanical and self-healing properties in these polymers are not correlated to the coordination dynamics unless the author can provide more convincing evidence.

=> The steric bulkiness of the counter anion can definitely impact the polymer network structure and mechanical properties. To address this issue, we conducted DFT calculation to obtain the energy-minimized structure of each counter anion, allowing us to estimate their volumes (**Fig. r4** and **Supplementary Table 2**).

Fig. r4 | van der Waals surfaces of the counter anions calculated from DFT (B3LYP/6-31G(d)) energy-minimized structure. a, acac⁻, b, OAc⁻, c, Cl⁻, d, OTf⁻, e, TFSI⁻.

Entry	Counter anion	van der Waals volume (Å ³)	van der Waals surface area (Å ²)
1	acac ⁻	93.56	124.89
2	OAc ⁻	55.31	75.34
3	Cl ⁻	20.41	36.46
4	OTf ⁻	81.92	107.36
5	TFSI ⁻	151.56	181.25

Supplementary Table 2. van der Waals volumes and surface areas of the counter anions calculated from DFT (B3LYP/6-31G(d)) energy-minimized structures.

=> We could find that TFSI⁻ anion is considerably larger compared to the other anions. This size effect is responsible for the observations that 1) the SAXS and SANS peaks of Zn-TFIS-BPy-PDMS near $q \sim 0.1 \text{ \AA}^{-1}$ appear in the higher q region in comparison to the other crosslinked polymers (**Supplementary Fig. 7b and d**), and 2) Zn-TFSI-BPy-PDMS exhibits lower mechanical strength yet higher stretchability compared to Zn-OTf-BPy-PDMS even though they are spectroscopically (film (**Supplementary Fig. 10**) and solution UV-vis spectra

(**Supplementary Fig. 3**) and rheologically (relaxation time spectra (**Fig. 3h** and **Supplementary Fig. 5b**)) almost identical. This means that OTf⁻ and TFSI⁻ anions are nearly identical in terms of the coordination dynamics and thus the differences in the mechanical properties are most likely due to the size effect. We also postulated that the absence of synergistic behavior with the combination of acac⁻ and TFSI⁻ is attributed from the steric bulkiness of TFSI⁻ anion (Main text, Page 12–13).

Main text, Page 7–8

When comparing Zn-OTf⁻ and Zn-TFSI-BPy-PDMS, it is reasonable to attribute the lower tensile strength and higher stretchability of Zn-TFSI-BPy-PDMS to the size of the anion (**Supplementary Table 2**). The bulkiness of TFSI⁻ likely leads to a notable plasticizing effect when incorporated into the polymer network.

=> The van der Waals volumes of the anions except for Cl⁻ (which is particularly small) and TFSI⁻ are comparable, and acac⁻ anion is even larger in size than OTf⁻ anion. Therefore, the mechanical and dynamic properties of these polymers cannot be solely attributed to a size-induced plasticizing effect. Many different factors should simultaneously affect the macroscopic properties; however, we believe that the coordination dynamics which means the kinetics of dynamic bond breakage/reformation is the major factor in accordance with our detailed studies on stress relaxation and rheological relaxation time spectrum analysis. While it is indeed true that the coordination dynamics is not the sole factor influencing the macroscopic behavior of the polymers, it is the primary factor for dynamic mechanical properties.

3) The author claimed that “the bridging coordination of acac⁻ and OAc⁻ are most likely to exist when two or more metal centers are in close contact.” It is true that acetate (OAc⁻) and

acetylacetonate (acac⁻) can bind to two metal ions in bridging mode, but that happens only when steric hindrance is not a problem (no bulky ligands are coordinated to the metal center). In this study, if pyridinyl ligands (especially when two bpy ligands) have coordinated to the metal center, I don't think acetate (OAc⁻) and acetylacetonate (acac⁻) can exist as bridging ligands.

=> We partially agree with the reviewer's opinion. It is true that bridging coordination is less viable under a sterically hindered environment. Nevertheless, there exist numerous metalloenzymes that possess sterically hindered active sites with bridged multinuclear complexes in the condensed protein structure. Here are some prominent examples: a) Dizinc leucine aminopeptidase (*Proc. Natl. Acad. Sci. USA* **96**, 11151–11155 (1999); *Coord. Chem. Rev.* **232**, 5–26 (2002)), b) Ribonucleotide reductase (*Science* **329**, 1526–1530 (2010)), c) Binuclear hydrogenase (*Science* **321**, 572–575 (2008); *Sci. Rep.* **10**, 10540 (2020)). The structures of those enzyme active sites consist of small ligands bridging two metal ions. Furthermore, there are several X-ray structures available that depict bridged synthetic dinuclear complexes of first-row transition metals with coordinated pyridyl (*Monatsh. Chem.* **151**, 543–547 (2020)) or bipyridyl (*Molecules* **24**, 3951 (2019); *J. Chem. Crystallogr.* **51**, 1–8 (2021)) ligands.

=> The formation of bridged polynuclear complexes is challenging but feasible when the ligands possess bridging capabilities and the metal ions are in close proximity. In the context of our polymer structure, the major component consists of hydrophobic PDMS. On the other hand, metal-ligand crosslinker exhibits greater polarity compared to the main polymer backbone due to the charged metal ion centers and counter anions. As a result of this polarity mismatch, the metal-rich domain would likely form, as described in the previous reports (*Macromolecules* **55**, 9126–9133 (2022); *Science* **358**, 502–505 (2017); *Polymers* **12**, 1680 (2020)). The presence of the metal-rich domain is consistent with the observed SAXS pattern,

which displays a single intense Bragg scattering peak. This peak indicates a domain-to-domain spacing of ~6 nm. Within this metal-rich domain, we could anticipate that the metal ions are in sufficiently close proximity for the formation of bridged dinuclear complexes under the applied stress.

=> To provide direct evidence of bridging coordination, we conducted X-ray absorption spectroscopy experiments. However, as previously noted, it was challenging to obtain meaningful results from these experiments. Due to the system's high complexity, particularly under strain, observing the bridging coordination directly within the polymer matrix proved to be challenging. Nevertheless, we were able to observe the presence of multiple energy dissipation modes in the cases of oxygen-based acac^- and OAc^- anions through the rheological relaxation time spectrum studies.

=> Following the reviewer's comment, we have revised the term 'bridging anion' to 'multimodal anion' because there is no direct experimental evidence of the 'bridging coordination'. We used the term 'multimodal' to highlight multiple coordination modes and multiple energy dissipation modes observed in our study. We thank the reviewer for pointing out what we overlooked.

Main text, Page 2, Abstract

Additionally, multimodal anions that can have diverse coordination modes provide unexpected dynamicity.

Main text, Page 4

We utilize three classes of counter anions (**Fig. 1b**): (i) non-coordinating anion, (ii) coordinating anion, and (iii) multimodal anion. A multimodal anion represents a coordinating anion that is capable of multiple coordination modes.

Main text, Page 8

Effect of diverse coordination modes

Main text, Page 8–9

We postulated that these intriguing mechanical properties might originate from the additional coordination modes of multimodal anions which can diversify the coordination environment (Fig. 2c). The hydrophobic nature of the PDMS backbone might exclude the charged metal centers, which in turn make them located close enough to have diverse coordination modes.

Fig. 1b | Design strategy for toughening self-healing polymer solely crosslinked by metal–ligand coordination. Impact of different types of anions on the coordination environment (top) and toughening of self-healing polymer through mixing of non-coordinating and multimodal anions (bottom).

Fig. 2c | Schematic illustration of counter anion effects on mechanical properties and energy dissipation mechanisms. Plausible energy dissipation mechanisms of Zn-X-BPy-PDMS polymers under mechanical stimuli; **c**, multimodal anions ($X = \text{OAc}^-$, acac^-).

4) Page 3, line 46-48, “Indeed, H-bonding motifs have a wider range of bonding modes and strength compared to other dynamic bonds such as metal–ligand coordination, π – π interaction, and dynamic covalent bond.” Actually, it is generally believed that metal–ligand coordination has wider range of bonding modes and strength compared to H-bonding.

=> We thank the reviewer for this suggestion and comment. Our intention was to convey that H-bonding exhibits a more dynamic and less directional nature compared to metal–ligand coordination. This is due to the fact that hydrogen bonding energies typically range from 1 to 40 kJ/mol (*ACS Omega* **6**, 9319–9333 (2021)), which is lower compared to the binding energies associated with metal-ligand interactions, ranging from 100 to 200 kJ/mol (*Coord. Chem. Rev.* **197**, 191–229 (2000); *Z. Kristallogr.* **228**, 311–317 (2013)). We revised the sentence as follows.

Main text, Page 3–4

Indeed, the combination of relatively weak hydrogen bonds, typically with binding energy ranging from 1 to 40 kJ/mol, allows the soft polymer network to have both toughness and dynamic behavior.²¹ Nevertheless, it is challenging to understand the detailed mechanisms for their dynamic mechanical properties due to the complicated binding dynamics of weak H-bonding. Among the other dynamic bonds, metal–ligand coordination of which the typical bond strengths ranges from 100 to 200 kJ/mol is of particular interest.^{11,22–27} Since it is more directional and predictable for bond formation compared to weaker H-bonding and its thermodynamic parameters possess rich tunability over a broad range, it would be a good model system to systematically understand the toughening mechanism of self-healing polymers.^{22,24}

5) The performance of polymers reported in this manuscript is not very competitive. In spite of their relative mechanical strength (about 1 MPa), they only self-heal at elevated temperature or for quite long time. Many better self-healing polymers have been reported previously.

=> It is true that there are several reported polymers which are self-healable under ambient condition with higher efficiency than our polymers (*Adv. Mater.* **32** 1903762 (2020)). However, it is important to note that developing a state-of-the-art self-healing polymer was not the primary focus of our research. Instead, our focus was on providing a detailed understanding of 1) the counter anion dependence on the mechanical and dynamic properties and 2) the synergistic effect between the counter anions and the toughening mechanism. Among the previous studies on self-healing polymers utilizing metal–ligand coordination as dynamic bonding, the strategies for the simultaneous improvement of seemingly conflicting mechanical and dynamic properties have involved 1) composite structures with other (usually inorganic) materials (*Nano Converg.* **6**, 29 (2019); *Eur. Polym. J.* **124**, 109448 (2020)), 2) in situ formation of inorganic clusters or minerals (*Science* **358**, 502–505 (2017); *Angew. Chem.* **132**,

5316–5321 (2020); *Nat. Commun.* **12**, 667 (2021)) and 3) synergy with other types of dynamic bonding (*ACS Appl. Mater. Interfaces* **9**, 28305–28318 (2017); *Chem. Eng. J.* **451**, 138673 (2023)). To the best of our knowledge, there is no report demonstrating the improvement in both mechanical (tensile strength) and dynamic (stretchability) properties solely based on the coordination dynamics observed in our research.

=> Indeed, the performance of the polymers, including mechanical, dynamic, and self-healing properties, is influenced by a combination of multiple factors. These factors encompass polymer backbone characteristics, the stereoelectronic effect of the ligand, the type of metal ion, its oxidation number, and so on. For the purpose of this study, we deliberately focused on a specific polymer system to elucidate the effect of counter anions, thereby emphasizing their impact on the overall performance. This approach allowed us to gain in-depth insights into the counter anion dependence of the properties, providing a proof-of-concept and facilitating detailed mechanistic investigations. Efforts are currently underway in our lab to apply this chemistry to the other polymer networks crosslinked by different metal–ligand interaction for 1) validating the universality of our strategy across diverse systems, and 2) developing polymers with better performance.

Reviewer #2

In their manuscript entitled "Toughening self-healing elastomer crosslinked by metal-ligand coordination through mixed counter anion dynamics", Park et al. present a intriguing study of the use of what they call bridging and non-coordinating ions to influence the mechanical properties of elastomers. Although the concept is interesting, and potentially adds new value to the growing literature on such materials, there are a number of items that should be addressed before publication

=> We appreciate the careful review by the reviewer. The reviewer commented that “*Park et al. present a intriguing study of the use of what they call bridging and non-coordinating ions to influence the mechanical properties of elastomers*”, but pointed out that there are “*a number of items that should be addressed before publication*”.

=> We have now carefully revised the manuscript, and our response to the reviewer’s helpful comments/suggestions are provided below. All main revisions are marked in **red font**.

1. Authors refer several times to the impact of their work on a "sustainable society" in the abstract and introduction. This is unclear and should be better explained or removed.

=> We thank the reviewer for this suggestion. We used the term “sustainable society” since intrinsically self-healing materials enable its iterative usage by effectively self-repairing damage, thus promoting sustainability in various applications. However, we found that the term “sustainable society” is more related to the environmental science. We have revised the term “sustainable society” to “**sustainable future**”, following the expression in the previously reported paper (*ACS Appl. Mater. Interfaces* **10**, 15331–15345 (2018)).

Main text, Page 3

Over recent decades, significant efforts have been devoted to developing intrinsically self-healing materials which can be readily repaired and used repeatedly upon mechanical damages for the realization of a sustainable future.¹⁻⁶

2. Lines 48-51. It is not obvious how the binding dynamics of H-bonding are more complicated than those of metal-ion complexes as described in this work. Please clarify.

=> We appreciate the reviewer's legitimate point. The reviewer #1 also raised the same question (comment #4), which prompted us to revise the main text. As detailed in our response to the reviewer #1's comment #4, we have emphasized the less directional and more dynamic nature

of H-bonding in comparison to metal–ligand coordination due to the typically weaker bond strength of H-bonding. This inherently dynamic nature of H-bonding adds complexity to the system, making it challenging to deeply study and understand.

=> We have provided our response to the reviewer #1's comment #4 below for convenience.

=> We thank the reviewer for this suggestion and comment. Our intention was to convey that H-bonding exhibits a more dynamic and less directional nature compared to metal–ligand coordination. This is due to the fact that hydrogen bonding energies typically range from 1 to 40 kJ/mol (*ACS Omega* **6**, 9319–9333 (2021)), which is lower compared to the binding energies associated with metal-ligand interactions, ranging from 100 to 200 kJ/mol (*Coord. Chem. Rev.* **197**, 191–229 (2000); *Z. Kristallogr.* **228**, 311–317 (2013)). We revised the sentence as follows.

Main text, Page 3–4

Indeed, the combination of relatively weak hydrogen bonds, typically with binding energy ranging from 1 to 40 kJ/mol, allows the soft polymer network to have both toughness and dynamic behavior.²¹ Nevertheless, it is challenging to understand the detailed mechanisms for their dynamic mechanical properties due to the complicated binding dynamics of weak H-bonding. Among the other dynamic bonds, metal–ligand coordination of which the typical bond strengths ranges from 100 to 200 kJ/mol is of particular interest.^{11,22-27} Since it is more directional and predictable for bond formation compared to weaker H-bonding and its thermodynamic parameters possess rich tunability over a broad range, it would be a good model system to systematically understand the toughening mechanism of self-healing polymers.^{22,24}

3. Line 55. There have been many studies of metal-ion coordinated binding in polymer networks, including studies of ion type, mixed ions, and of the chemical environment, notably by Holten-Anderson, among others. A google scholar search of metal ion coordination in

elastomers returns over 20,000 results. Authors should be much clearer about how their results provide substantially new information, and should be careful of claims of novelty given the large literature.

=> It is indeed true that there have been many reports exploring the effect of coordination thermodynamics and kinetics on the macroscopic properties of metal–ligand crosslinked soft polymer network. We have diligently cited relevant literatures in our manuscript. What sets our study apart is the remarkable achievement of simultaneously improving both mechanical properties (i.e. tensile strength), and dynamic properties (i.e. stretchability and self-healing efficiency) of self-healing polymer solely by controlling the coordination equilibrium and dynamics through counter anions. Although many previous investigations have contributed to the understanding of the metal–ligand crosslinked polymer systems, there is a lack of existing reports demonstrating significant mechanical toughening by fine-tuning the coordination dynamics.

=> As the reviewer mentioned, the outstanding works have been conducted by Prof. Holten-Anderson, particularly regarding mussel-inspired iron catecholate crosslinking. We have also adopted a similar method to what he used for the relaxation time spectrum ($H(\tau)$) analysis. In terms of mechanical toughening, however, it is important to note that our study stands out from existing reports. To the best of our knowledge, there is no previous study demonstrating a strategy solely based on the coordination dynamics that leads to simultaneous enhancement in both mechanical and dynamic properties. This highlights the novelty and significance of our findings.

=> We agree to the reviewer that the sentence in Line 55 may have been misleading. To clarify, we have revised the sentence as follows.

Main text, Page 4

Although a number of previous investigations have contributed to the understanding of the

metal–ligand crosslinked self-healing polymers, there is a lack of reports demonstrating significant mechanical toughening through fine-tuning the metal–ligand coordination environments (**Fig. 1a**).

4. Lines 70-75. Claims here are vague. Define "extra dynamicity", "synergistic actions" and what is meant by "universal toughening" given the limited material set tested here.

=> We appreciate the reviewer for pointing out the presence of vague terms in our manuscript. We made modifications to eliminate those vague terminologies and provide clearer description.

Main text, Page 4–5

Notably, multimodal anions have additional coordination modes that provide unanticipatedly high stretchability despite the strong coordination of the anion. More intriguingly, the synergy of non-coordinating and multimodal anions simultaneously enhances the mechanical toughness and self-healing efficiency. This demonstrates a new toughening mechanism of self-healing polymer with metal-ligand coordination bonds.

=> We also acknowledge the reviewer's concern regarding the claim of "universal toughening" in our study. We have removed all the words "universal" or "universality" from the manuscript and made appropriate modifications to accurately represent the scope of our research.

Main text, Page 13–14

Counter anion effects on the Cu²⁺-crosslinked system

To confirm the applicability of anion-dependent coordination dynamics to the system with other kinetically labile transition metal ions, we prepared the polymers crosslinked by Cu²⁺–BPy coordination bonds (Cu-X-BPy-PDMS) and investigated whether there are differences in mechanical properties depending on the counter anion or not (**Fig. 5a and 5b**).

5. Line 83. What is the evidence of tris coordination? Is there a distinct spectroscopic signal that you can provide?

=> We thank the reviewer for raising this valid point, which was also mentioned by the reviewer #1 in his/her comment #1. As described in our detailed response to the reviewer #1's comment, we provided spectroscopic evidences to support the predominant formation of the tris(BPy)zinc(II) complex (**Supplementary Figs. 2, 3, and 8–10**), even in the presence of coordinating counter anions. We also attempted to directly observe it by X-ray absorption spectroscopy, but the results were not clear due to the stability issue of the polymer films under X-ray exposure.

=> We have modified the term “tris(BPy)zinc(II) coordination complexes” to “**Zn²⁺-BPy coordination**”, and “tris(BPy)zinc(II) crosslinkers” to “**the crosslinkers**”.

=> We have provided a part of our response to the reviewer #1's comment #1 below for convenience.

=> Although the coordinating counter anions (acac⁻, OAc⁻, and Cl⁻) participate in the coordination sphere, the film UV-vis spectra (**Supplementary Fig. 10**) indicate that tris(BPy)zinc(II) coordination complexes are dominant. The major absorption peaks (200 ~ 350 nm) are similar to those of non-coordinating anion (OTf⁻ and TFSI⁻), and the absorbance of the LLCT band (near 400 nm) is less than 10% of the absorbance of the highest peak (acac⁻; 10%, OAc⁻; 7%, Cl⁻; 10%). In addition, the solid state ¹³C CP/MAS NMR spectra look almost identical except for the peak broadness for all the counter anions (**Supplementary Fig. 8**). Furthermore, the overall scattering profiles obtained from SAXS, WAXS and SANS experiments (**Fig. r1**) are almost superimposable for all the anions, indicating that the crosslinked polymer networks are similar at 0% strain regardless of the counter anion.

Fig. r1 | Overall a, SAXS, WAXS and b, SANS scattering patterns of Zn-X-BPy-PDMS films.

=> To obtain direct experimental evidence, we performed X-ray absorption spectroscopy (XAS) of the Zn^{2+} -BPy crosslinked polymer films, which are expected to reveal information about the local environment around the metal center (**Fig. r2**). We could find small differences according to the counter anions in XANES (X-ray Absorption Near Edge Structure) and EXAFS (Extended X-ray Absorption Fine Structure), however, there was a stability issue. The crosslinked films turned black within a minute of X-ray exposure. This X-ray beam damage makes it hard to ensure that the differences in the spectra come from the structural differences around the crosslinker.

Fig. r2 | XANES and EXAFS spectra of Zn-acac-, Zn-Cl-, and Zn-OTf-BPy-PDMS films.

⇒ We also performed solution UV-vis studies to further support our proposed mechanism (**Supplementary Figs. 2 and 3**). Due to the solubility difference of BPy-PDMS polymer and zinc salts, it was unable to try Job's analysis. Therefore, we conducted titration of a CHCl_3 solution of BPy-PDMS ($33 \mu\text{M}$) by continuously increasing the molar ratio of Zn^{2+} salt solutions (4.95 mM , solvent: EtOH for $\text{Zn}(\text{acac})_2$ and MeOH for the others). Regardless of the counter anion, no significant difference was observed up to the addition of 0.33 equivalent zinc salt (**Supplementary Fig. 2a, c, e, and Supplementary Fig. 3a, c**). We could observe well-defined isosbestic points for all the cases, which means the predominant formation of $\text{tris}(\text{BPy})\text{Zn}(\text{II})$ complex. The counter anion dependence is observed when the additions are above 0.33 equivalent. As the molar ratio of Zn^{2+} to BPy ligand exceeds 1:3, the isosbestic points for all the cases move to shorter wavelength, indicating the formation of Zn^{2+} complexes with different binding stoichiometry. For the non-coordinating counter anions (OTf^- and TFSI^-), the spectral changes above 0.33 eq are almost identical (**Supplementary Fig. 3b and d**). On the other hand, the spectra of coordinating anions (acac^- , OAc^- , and Cl^-) above 0.33 eq exhibit an apparent counter anion dependence (**Supplementary Fig. 2b, d, and f**), presumably

by the coordination of anions. For the case of acac^- , the emergence of absorption band under 300 nm with increasing amount of Zn^{2+} salt (**Supplementary Fig. 2a and b**) might be due to the absorption of free $\text{Zn}(\text{acac})_2$ salt (**Fig. r3**). It is true that the highly diluted solution state cannot represent the situation in the condensed phase. Combining the results from both film and solution UV-vis studies, however, we believe that the dominant coordination complex under the 1:3 ratio of Zn^{2+} to BPy is the tris(BPy)zinc(II) complex regardless of the binding ability of the counter anion.

Supplementary Fig. 2. UV-vis absorption spectra obtained by titration of a CHCl_3 solution of BPy-PDMS (33 μM) with Zn^{2+} salt solutions of coordinating counter anions. **a, b**, $\text{Zn}(\text{acac})_2$, **c, d**, $\text{Zn}(\text{OAc})_2$, **e, f**, ZnCl_2 .

Supplementary Fig. 3. UV-vis absorption spectra obtained by titration of a CHCl₃ solution of BPy-PDMS (33 μM) with Zn²⁺ salt solutions of non-coordinating counter anions. **a, b,** Zn(OTf)₂, **c, d,** Zn(TFSI)₂.

Fig. r3 | Normalized absorption spectrum of Zn(acac)₂ salt (33 μM) in CHCl₃/EtOH mixed

solvent (150:1 v/v).

Main text, Page 5

The predominant formation of Zn²⁺-BPy complexes was confirmed by solution UV-vis titration studies (Supplementary Figs. 2 and 3).

6. In the mechanical analysis, starting on ~Line 99, it is unclear how the authors selected the strain rates for tensile testing, and the extent to which the report parameters may depend on strain. Loading rates of 100% per minute are reported, which appear to be extremely fast, and may engage the fast relaxation time scales reported later.

=> We respectfully disagree with the reviewer's point. The loading rate of 100% min⁻¹ for tensile testing is a conventional and commonly used setting, and even faster loading rates have often been applied for highly dynamic PDMS-based soft polymers. We chose a loading rate of 100% min⁻¹ with the specific purpose of effectively demonstrating the impact of stress relaxation dynamics in the tensile test. As shown in **Fig. 3a**, the stress-strain curves of Zn-OTf-BPy-PDMS and Zn-TFSI-BPy-PDMS exhibit a steady plateau right after the elastic region. This denotes the rapid and effective stress relaxation of the polymers containing non-coordinating anions, which is consistent with our shear stress relaxation studies. In contrast, the stress-strain curves of the other polymers with slower relaxation dynamics depict a gradual increase in stress until fracture occurs. Our intention was to highlight this notable difference in the tensile profiles.

=> As pointed out by the reviewer, the stress relaxation studies (**Fig. 3b**) indicate that a considerable amount of time is necessary for the polymers to fully relax the applied stress. However, it is important to note that direct correlation between the time scale of the relaxation tests and the results from the tensile tests should not be done. There are two reasons. First, the

stress relaxation experiments were performed under a shear strain of 3% to obtain clear information about the behavior of the polymer network. Since the type of the applied force is fundamentally different from the tensile tests, it is not appropriate to directly correlate the time scale of shear stress relaxation with the results obtained from the tensile measurements. Secondly, the relaxation time is dramatically reduced as the applied strain increases. This can be attributed to the deformation of the polymer networks under higher strain, which leads to significantly faster stress relaxation (*Macromolecules* **44**, 3988–4000 (2011)). The reason that we conducted the shear stress relaxation studies under a low strain (within the linear viscoelastic region) was to compare the intrinsic dynamic properties of the polymer films without the influence of the permanent deformation. Therefore, this stress relaxation time scale is specifically applicable to the elastic region of the material, and may not be representative of the entire tensile stress-strain curves.

Do what extent are the elastic modulus, tensile strength and fracture strain dependent on the strain rate? Can the differences in anion type be attributed to differences in strain rate relative to the intrinsic relaxation time scales of the system?

=> For viscoelastic materials, the strain rate dependence is one parameter that demonstrates the viscoelasticity. When a material is more elastic and solid-like, the effect of the strain rate tends to be less pronounced. In contrast, when a material exhibits higher viscoelasticity and more liquid-like properties, the mechanical properties are more dependent on the strain rate. In general, viscoelastic materials show larger fracture strain and lower stress level at slower strain rates. This is because the slower strain rate allows the material more time to relax the applied stress and accommodate the deformation.

=> We conducted the measurements of stress–strain curves with newly prepared polymer films at various loading rates (50% min⁻¹, 100% min⁻¹, and 500% min⁻¹). All the polymers follow

the general trend (**Fig. r5**) where larger fracture strain and lower stress level are observed at slower strain rates. However, the influence of the strain rate is highly dependent on the counter anion. The polymers with stronger anion binding and slower stress relaxation dynamics show less pronounced effect of the strain rate on the mechanical properties. Specifically, in the case of Zn-acac-BPy-PDMS, the strong binding of the bidentate anion results in the stress–strain curves that are almost superimposable regardless of the strain rate. In stark contrast, the polymers with non-coordinating anions exhibit strong dependence on the strain rate. This result aligns with the other studies which explore the polymer viscoelasticity, including cyclic loading tests, rheological frequency sweeps, and shear stress relaxation tests (**Fig. 3**).

Fig. r5 | Stress–strain curves of Zn-X-BPy-PDMS polymer films with a sample width of 3 mm, a thickness of 0.2–0.4 mm, and a length of 5 mm at loading rates of 2.5, 5, and 25 mm min⁻¹. **a**, acac⁻, **b**, OAc⁻, **c**, Cl⁻, **d**, OTf⁻, **e**, TFSI⁻.

Can authors compare the expected differences in energetic bonding given the anions selected?

=> As we discussed in the main text, the order of coordination ability is $acac^- > OAc^- > Cl^- > OTf^- > TFSI^-$. It is derived from both thermodynamics and kinetics of the bond formation. In terms of thermodynamics, the stereoelectronic factor of the anion needs to be considered. The non-coordinating anions, OTf^- and $TFSI^-$, have electron-deficient oxygen and nitrogen atoms as a result of the strong electron-withdrawing effect of CF_3 functional groups. This electronic effect hinders the coordination of these anions. In addition, the steric bulkiness of $TFSI^-$ anion induces significant steric hindrance with the ligands when it is coordinated to the metal center, thus leading to even weaker binding of this anion. The coordinating anions have lone pair electrons which can bind to the metal center. According to the ligand field theory and the spectrochemical series, Cl^- anion is classified as a weak field ligand due to its strong π -donation. OAc^- and $acac^-$ anions are closer to strong field ligands compared to Cl^- . Among them, $acac^-$ anion is particularly strong due to the chelate effect. It is related to the kinetic effect. Since $acac^-$ anion functions as a bidentate chelate, the second metal–oxygen bond formation is considerably facilitated by the first metal–oxygen coordination thanks to the enhanced collision frequency. We found that the difference in the coordinating ability of the anions is the primary factor that determines the mechanical properties. The involvement of the anion in the coordination sphere results in slower ligand exchange dynamics and, consequently, slower stress relaxation (**Fig. 2 and 3b**). This slower relaxation process contributes to an increased elasticity of the polymer.

7. Line 127. Define stretchability and relate this to the terms presented earlier (in particular, fracture strain). Explain more clearly how you relate this to toughness and dissipation.

=> The term “stretchability” means the ability of a material to be stretched without experiencing material failure. In the case of unidirectional tensile test, “stretchability” is

identical to “fracture strain”, which represents the maximum elongation the material can withstand before it breaks. “Toughness” represents the amount of absorbed energy before it breaks, which represents the area under the stress–strain curve from zero strain to “fracture strain”. Therefore, to enhance the material toughness, it is crucial to improve both mechanical strength and stretchability.

=> "Energy dissipation" refers to the process by which a material dissipates or loses energy when subjected to an applied load. It is closely related to the dynamic or viscoelastic behavior of the material during deformation, as it involves the conversion of mechanical energy into other forms, such as 1) frictional heat derived from the polymer chain movement or 2) release of physical/chemical energy stored in the physical/chemical bonding within the polymer network. The loss through frictional heat is related to the dynamicity or mobility of the polymer chain. If the polymer has higher mobility (lower T_g), the polymer chains interact more with each other, resulting in increased heat generation during deformation. The release of bond energy depends on both thermodynamics and kinetics of the bond formation. If the bonding is dynamic and can be reformed reversibly, the polymer loses energy by bond breaking and recovers it through subsequent bond reformation. This recovered dynamic bonding can then dissipate energy again. Faster bond breaking/reformation dynamics or kinetics enhance the ability of the material to dissipate the applied energy. In general, polymers with high “stretchability” exhibit enhanced energy dissipation capabilities. In our study, we used the polymers with the same PDMS backbone yet different coordination dynamics. Therefore, the dynamics of reversible metal–ligand bond formation are the primary factor that affects energy dissipation or stress relaxation.

=> We found that the cited reference paper regarding to this relationship between energy dissipation and coordination dynamics is inappropriate. We replaced it to more relevant paper. We thank the reviewer for bringing this issue to our attention.

Main text, Page 7

The capability of energy dissipation relies on the bond-breaking-reformation dynamics of M–BPy coordination.³⁷

References

37 Wanasinghe, S. V., Dodo, O. J. & Konkolewicz, D. Dynamic Bonds: Adaptable Timescales for Responsive Materials. *Angew. Chem. Int. Ed.* **61**, e202206938 (2022).

=> All three terms mentioned by the reviewer are widely accepted in the fields of polymer physics and chemistry, particularly in relation to viscoelastic polymers. In short, “stretchability” and “energy dissipation” are associated with the viscous (or dynamic) behavior of the polymer, and “toughness” encompasses both elastic and viscous properties.

8. Lines 127-129. Authors suggest that kinetic competition between diffusion and a coordination reaction may play a role in network dynamics -- this should be expanded and clarified.

=> The ligand BPy is embedded in the polymer backbone. Thus, the entire polymer chain needs to move for coordination with the metal ions. In contrast, small counter anions are relatively free to move within the polymer matrix. This difference in mobility between the ligand and the counter anions can result in a remarkable difference in their likelihood of encountering the metal ion.

=> The hydrophobic nature of the PDMS backbone might exclude polar charged species, resulting in the formation of metal-rich polar domains, as we detailed in our response to the reviewer #1's comment #3. Since the BPy ligand is charge-neutral, the coulombic interaction between the metal ions and the counter anions leads to a significantly higher local concentration of the counter anions around the metal ions. This can greatly increase the collision frequency

between the counter anions and the metal ions, thus enabling their coordination during deformation.

=> We thank the reviewer for this suggestion. We have made modifications accordingly.

Main text, Page 8

Although the coordinating ability of the bidentate BPy ligand is superior to the monodentate coordinating anion, higher mobility of the small counter anions compared to the BPy ligands embedded in the polymer backbone enables its coordination with the metal. In addition, the counter anions would have a higher collision frequency with the metal ions due to its higher local concentration which is achieved by coulombic interactions.

9. Lines 197-200. Authors should provide more information about the optical methods used, as well as the rheometry measurements. It is difficult to understand the argument as written.

=> For the optical microscope images of the polymer films, we prepared the samples as illustrated in **Fig. r6**. The polymer film was placed on a glass slide, and we captured an optical microscope image of the pristine film. Subsequently, we damaged the film by gently cutting it with a razor blade. This damaged film was subjected to a temperature of 50 °C for 6 h in an oven, and another image of the film was taken. As illustrated in **Fig. 4c**, only Zn-acac-OTf-BPy-PDMS film exhibited complete healing, despite the largest deformation. However, this method provides only indirect comparison of the self-healing efficiency, as the damaging process is entirely dependent on arbitrary force applied manually. Therefore, we conducted temperature-dependent self-healing test with the rheometer to quantitatively compare the self-healing efficiency.

Fig. r6 | Sample preparation for optical microscope images.

=> The detailed experimental setup for this temperature-dependent self-healing test is depicted in the inset illustration of **Supplementary Fig. 14** and described in the **Methods** part. We adopted the previously reported method by Prof. Takuzo Aida and Prof. Laurent Corté (*Science* **359**, 72–76 (2018); *J. Am. Chem. Soc.* **143**, 15279–15285 (2021); *Soft Matter* **8**, 1681–1687 (2012)) with slight modifications. At high temperatures, the movement of polymer chains is facilitated due to increased entropic contribution. In general, this enhanced chain mobility promotes 1) the reformation of dynamic bonds by increasing the collision frequency between dynamic bonding units, and 2) the interfacial diffusion of polymer chains at the self-healed interface. As a result, the self-healing efficiency increases with increasing temperature as depicted in **Supplementary Fig. 14**. Under identical measurement conditions, we utilized two experimentally observed values to compare the self-healing efficiency: 1) the temperature at which self-healing starts, and 2) the recovery of the intrinsic mechanical strength. For both of these factors, the Zn-acac-OTf-BPy-PDMS film shows the highest self-healing efficiency among the other polymer films as described in **Main text, Page 12**.

=> For clearer description of the self-healing test, we have revised the main text as follows.

Main text, Page 11

Optical microscope images showed that the Zn-acac-OTf-BPy-PDMS film damaged by a razor blade underwent complete healing after being heated at 50 °C for 6 h, while the other films were not successful in healing under identical conditions (**Fig. 4c**). To quantitatively compare the self-healing efficiency, we measured temperature-dependent healing behavior by using a rheometer according to the previously reported method (see **Methods** for details on the procedure).^{13,49}

10. Line 215. The evidence for the role diversification of the coordination environment is not unclear. Is this conjecture based on observations, or is there specific evidence of difference here? Authors should be clearer about delineating results and interpretation.

=> We thank the reviewer for this suggestion. The diversification of the coordination environment under the applied strain directly translates to the diversification of the energy dissipation modes, since the energy dissipation in our system relies on the coordination dynamics. The number of energy dissipation modes and how much energy dissipates for each mode can be estimated through the analysis of the relaxation time spectrum ($H(\tau)$), as this experimental information indicates the amount of energy dissipated at the given relaxation time (τ).

=> Since the analysis on the relaxation time spectrum is more directly related to the diversification of coordination (or energy dissipation) modes, we have made a revision as follows.

Main text, Page 12

On the other hand, the relaxation time spectrum obtained from the master curve of Zn-acac-OTf-BPy-PDMS shows broad and multiple peaks similar to that of Zn-acac-BPy-PDMS (**Fig. 3e** and **Supplementary Fig. 17a**), which suggests the diverse coordination environment derived from acac⁻ anion. The contribution of acac⁻ was also detected by the development of

an absorption band around 400 nm with the increasing equivalent of acac^- (**Supplementary Fig. 13**).

=> We agree with the reviewer's claim regarding the need for a clearer description. Accordingly, we have put more details on the revised plots regarding to the relaxation time spectrum. We have added **log-normal relaxation spectra** for each energy dissipation mode as dashed lines to clearly visualize and highlight the individual contributions of each mode in the overall energy dissipation process. We hope that these modifications provide a clearer understanding of our findings.

Fig. 3f-h | bottom, Relaxation time spectra, $H(\tau)$ of **f**, Zn-acac-BPy-PDMS, **g**, Zn-Cl-BPy-PDMS, and **h**, Zn-OTf-BPy-PDMS. The dashed lines in the plot indicate the spectra of each energy dissipation mode.

Fig. 4e | bottom, Relaxation time spectrum, $H(\tau)$ of Zn-acac-OTf-BPy-PDMS which exhibits acac⁻-like broad and multiple peaks. The dashed lines in the plot indicate the spectra of each energy dissipation mode.

Fig. 5d-f | bottom, Relaxation time spectra, $H(\tau)$ of **d**, Cu-acac-BPy-PDMS, **e**, Cu-OTf-BPy-PDMS, and **f**, Cu-acac-OTf-BPy-PDMS. The dashed lines in the plot indicate the spectra of each energy dissipation mode.

Supplementary Fig. 5. bottom, Relaxation time spectra, $H(\tau)$ of **a**, Zn-OAc-BPy-PDMS, **b**, Zn-TFSI-BPy-PDMS. The dashed lines in the plot indicate the spectra of each energy dissipation mode.

Supplementary Fig. 16. c, bottom, Relaxation time spectrum, $H(\tau)$ of Zn-acac-TFSI-BPy-PDMS which exhibits $acac^-$ -like broad and multiple peaks. The dashed lines in the plot indicate

the spectra of each energy dissipation mode.

11. Claims of universality are not supported since only one polymer system has been tested.

This should be rephrased.

=> We thank the reviewer for this suggestion. Accordingly, we have removed all the words "universal" or "universality" from the manuscript and made appropriate modifications to accurately represent the scope of our research.

Main text, Page 13–14

Counter anion effects on the Cu²⁺-crosslinked system

To confirm the applicability of anion-dependent coordination dynamics to the system with other kinetically labile transition metal ions, we prepared the polymers crosslinked by Cu²⁺-BPy coordination bonds (Cu-X-BPy-PDMS) and investigated whether there are differences in mechanical properties depending on the counter anion or not (**Fig. 5a and 5b**).

12. Were the complexes and materials sensitive to water or the presence of other solvents?

=> The majority of the polymers in our study are composed of highly hydrophobic PDMS chains. While the metal–ligand crosslinker may be influenced by water binding, the overall hydrophobic nature of the polymer films ensures their stability in water. We conducted contact angle measurements using water to confirm the hydrophobicity of the polymer films (**Supplementary Fig. 21**). Regardless of the counter anion, the contact angle values were consistently over 100°, indicating great hydrophobicity of the films.

Supplementary Fig. 21. Sessile drop contact angle measurement of Zn-X-BPy-PDMS polymer films. X = a, acac^- (107.8°), b, OAc^- (109.7°), c, Cl^- (102.0°), d, OTf^- (109.7°), e, TFSI^- (109.7°). The results indicate the hydrophobic nature of the PDMS-based polymer films.

=> Since the polymers are not covalently crosslinked, they are soluble in certain non-polar organic solvents such as dichloromethane, chloroform, toluene. When exposed to these solvents, the polymers undergo a process of swelling, and eventually dissolve completely as the solvent molecules solvate the polymer chain.

13. Figure 1: In panel B it is not clear what authors mean by "dynamics" and how they judge fast versus slow. In particular, panels (B i) and (B ii) are identical except for the arrow direction -- how does directionality factor into descriptions of dynamics? Also, why is (B iii) considered super-slow? And, how are any of these considered tris coordinated? In the lowest panel, right hand side, how are any of the bridging anions playing a role (red box)?

=> We appreciate the reviewer's valid point. The term 'dynamics' specifically refers to the kinetic information of the ligand exchange process (**Fig. 2**). As the reviewer pointed out, we understand that it is not appropriate for the thermodynamic equilibrium of the counter anion binding as depicted in the top part of panel b in **Fig. 1**, since the thermodynamic equilibrium is

indeed related to the coordinating ability of the counter anions. We are grateful to the reviewer for pointing out this oversight. Accordingly, we have removed the words related to ‘dynamics’ in **Fig. 1 and its caption**.

=> The arrow direction in panel b indicates the direction of thermodynamic equilibrium of the counter anion coordination. For non-coordinating anions, the lack of coordinating ability thereof makes the formation of anion-bound metal complex less viable. On the other hand, the coordinating anions can stabilize the metal center by energetically favorable formation of coordination bonds when there are vacant coordination sites. As illustrated in **Fig. 2**, this formation of anion-bound coordination complexes during the ligand exchange process can impede the energy dissipating ligand exchange. For **multimodal anions (initially bridging anions; we have modified this term for clarity)**, the anions are able to form additional coordination modes, which provide further stabilization through the anion-bound forms. The higher denticity of the multimodal anions is also responsible for super-slow ligand exchange process.

=> As we mentioned above, we agree that this anion binding equilibrium is related to the thermodynamic aspect of the process, and not related to the ‘dynamics’ of the ligand exchange process. The effect of ‘dynamics’ is illustrated in **Fig. 2**; thus, modifications have made with **Fig. 1** to prevent misunderstanding.

=> For the dominant formation of the tris coordinated complex, we have provided UV-vis and NMR spectroscopic evidences (**Supplementary Figs. 2, 3, and 8–10**), and SAXS and SANS scattering information (**Supplementary Fig. 7 and Fig. r1**). For more details, please see our response to the comment #5.

=> In the bottom part of **Fig. 1, panel b**, our intention was to provide a simplified and schematic illustration of the coexistence of non-coordinating and multimodal anions within the polymer network of the mixed anion system. We demonstrated throughout our mechanistic studies that

these different types of anions function independently within the polymer network, eliciting a synergistic toughening effect.

Fig. 1b | Design strategy for toughening self-healing polymer solely crosslinked by metal–ligand coordination. Impact of different types of anions on the coordination environment (top) and toughening of self-healing polymer through mixing of non-coordinating and multimodal anions (bottom).

14. Figure 2: In panel B, why are the anion coordination and slow exchange transitions irreversible? (same question for panel C). Can Raman spectroscopy or imaging determine any of these transitions or complexes more quantitatively?

=> Indeed, when anions possess adequate coordinating ability and there are vacant coordination sites, they can undergo reversible coordination as they are not a part of the polymer backbone and located in close proximity to the metal center. We appreciate the reviewer for pointing out this oversight. We have revised **Figs. 2b and 2c** to accurately depict the anion binding equilibrium as shown in the scheme of **Fig. 1b**.

=> For the ligand exchange transition, we agree that the association-dissociation process itself is reversible. However, it is important to consider that the polymer film is under uniaxial strain. In our system, the ligand BPy is embedded in the polymer backbone. When the polymer is subjected to tension, the polymer chain reorganization and the ligand exchange should be restricted. The initial response to the applied strain involves the dissociation of the M–BPy bonds, followed by the reassociation with the available BPy ligand. Therefore, this sequential process renders the overall ligand exchange process irreversible under the strain.

Fig. 2 | Schematic illustration of hypothesized counter anion effects on mechanical properties and energy dissipation mechanisms. Plausible energy dissipation mechanisms of

Zn-X-BPy-PDMS polymers under mechanical stimuli; **a**, Non-coordinating anions ($X = \text{OTf}^-$, TFSI^-), **b**, coordinating anions ($X = \text{Cl}^-$), and **c**, multimodal anions ($X = \text{OAc}^-$, acac^-).

=> Following the reviewer's valuable suggestion, we conducted Raman (**Fig. r7**) and FT-IR (**Fig. r8**) spectroscopic studies. However, both spectroscopic studies did not reveal any significant variation associated with the type of the anion. In the Raman spectra, we could observe three major peaks at Raman shifts of 470, 690, and 1590 cm^{-1} . Through control experiments using non-crosslinked BPy-PDMS and bis(aminopropyl) terminated PDMS, we found that the peaks at 470 and 690 cm^{-1} originated from the PDMS backbone, and the peak at 1590 cm^{-1} corresponded to the amide carbonyl of the BPy unit. The position and intensity of the major peaks were almost identical for all the crosslinked polymer samples, and it was challenging to extract meaningful information from the minor peaks due to the fluorescence background originating from the Zn^{2+} -BPy coordination complexes. When we subjected Zn-acac-, Zn-Cl-, and Zn-OTf-BPy-PDMS polymers to a 50% strain, we observed only a slight development of minor peaks in 1250–1500 cm^{-1} region (**Fig. r7b–d**). However, it was difficult to assign them due to the broadened nature. While more peaks were observed in FT-IR spectra compared to the Raman spectra (**Fig. r8**), the significant broadening of the characteristic stretching peaks hinders the detailed and conclusive analysis.

=> Additionally, we conducted X-ray absorption spectroscopy, EXAFS and XANES (**Fig. r2**), to obtain direct evidence of the local coordination environment. However, we encountered challenges due to the unstable nature of the polymer films under X-ray irradiation. Considering the inconclusive results of our spectroscopic studies, we have concluded that more reasonable approach to elucidate the mechanism is through the analysis of the dynamic mechanical behavior, such as shear stress relaxation, rheological frequency sweeps, and relaxation time spectra analysis.

Fig. r7 | **a**, Raman spectra of the polymer films: in order from top to bottom, bis(3-aminopropyl) terminated PDMS, BPy-PDMS, Zn-TFSI-BPy-PDMS, Zn-OTf-BPy-PDMS, Zn-Cl-BPy-PDMS, Zn-OAc-BPy-PDMS, and Zn-acac-BPy-PDMS. **b–d**, Comparison of Raman spectra before and after 50% stretching of **b**, Zn-acac-BPy-PDMS, **c**, Zn-Cl-BPy-PDMS, and **d**, Zn-OTf-BPy-PDMS.

Fig. r8 | FT-IR spectra of Zn-acac-BPy-PDMS (red), Zn-Cl-BPy-PDMS (gray), Zn-OTf-BPy-PDMS (blue), and BPy-PDMS (black).

15. Figure 3: In panel A, how repeatable are these curves? How many replicates were performed? In panel C, what was the effect of loading rate, and waiting time before reloading? Did you see a change in residual strain when loading dynamics were altered?

=> As mentioned in the **'Methods'** section, at least three samples are tested for each polymer film, and the most reproducible curve was chosen. **Fig. r9** presents the multiple strain–stress curves of Zn-X-BPy-PDMS polymers obtained at a loading rate of 100% min⁻¹. For Zn-Cl-BPy-PDMS, we found that there was a critical error during the measurement of the curve. Therefore, we remeasured it and repeated the measurement for five times to ensure the accuracy.

Fig. r9 | Stress–strain curves of Zn-X-BPy-PDMS polymer films with a sample width of 3 mm, a thickness of 0.2–0.4 mm, and a length of 5 mm at a loading rate of 5 mm min⁻¹. At least three samples were tested for each polymer film. a, acac⁻, b, OAc⁻, c, Cl⁻, d, OTf⁻, e, TFSI⁻.

Fig. 3a | Effects of counter anions on mechanical and dynamic properties of Zn-X-BPy-PDMS. Stress–strain curves of Zn-X-BPy-PDMS polymer films (acac⁻; red, OAc⁻; orange, Cl⁻; gray, OTf⁻; blue, TFSI⁻; green) with a sample width of 3 mm, a thickness of 0.2–0.4 mm, and a length of 5 mm at a loading rate of 5 mm min⁻¹.

=> For the cyclic loading–unloading tests, the cyclic measurements were iterated three times from 0 to 50% strain with a time interval of 30 mins between cycles. As shown in the newly added **Supplementary Fig. 20** and the left panel of **Fig. r10**, the second and third cycles exhibited shift of the stress–strain profile to a lower stress level. However, it is important to note that the residual strain and maximum tensile strength remain almost identical for all the polymers, regardless of the number of cycles performed. In order to enhance clarity in the main figure (**Fig. 3c**), we have included the stress-strain profiles of only the first cycle. However, for a comprehensive understanding of the cyclic behavior, we have provided the stress-strain profiles for all three cycles in **Supplementary Fig. 20**.

=> The effect of the strain rate to these cyclic profiles was also examined (**Fig. r10**). The cyclic tests were performed under loading rates of 50, 200, and 500% min⁻¹. Before presenting accounts for the influence of the strain rate, we apologize for the poor quality of the newly measured cyclic curves (at strain rates of 50 and 500% min⁻¹) due to an issue with our load cell. We initially used 5 N load cell to clearly depict the results from the tensile tests, however, the newly measured curves were obtained from 100 N load cell because of malfunctioning 5 N load cell. Despite the poor quality, we observed a consistent trend in the strain rate dependence, which aligns with the strain rate-dependent stress-strain curve measurements. Specifically, at higher strain rates, we observed an increase in mechanical strength and a decrease in residual strain. This can be attributed to the insufficient time at higher strain rates for polymer backbone reorganization and energy dissipation. In comparison to the non-coordinating anions, the polymer network having the anions with coordinating ability exhibits higher elasticity, and its mechanical properties, including residual strain, are less influenced by the strain rate, which aligns well with the other tensile and rheological tests. The cyclic profiles of Zn-OTf-BPy-PDMS at strain rates of 50 and 500% min⁻¹ appear as outliers presumably due to the imprecise measurement at low force values thereof.

Fig. r10 | Stress–strain curves of the polymer films with a sample width of 3 mm, a thickness of 0.2–0.4 mm, and a length of 5 mm under cyclic loading at loading rates of 10 (left), 2.5 (center), and 25 (right) mm min^{-1} . Three cycles with a time interval of 30 min were performed for each measurement. **a, acac^- , **b**, OAc^- , **c**, Cl^- , **d**, OTf^- , **e**, TFSI^- .**

Supplementary Fig. 20. Stress–strain curves of the polymer films with a sample width of 3 mm, a thickness of 0.2–0.4 mm, and a length of 5 mm under cyclic loading at loading rates of 10 mm min^{-1} . Three cycles with a time interval of 30 min were performed for each measurement. **a**, acac^- , **b**, OAc^- , **c**, Cl^- , **d**, OTf^- , **e**, TFSI^- .

Reviewer #3

This manuscript brings insightful design strategy for self healing polymers using metal-ligand coordination. Overall the manuscript is in the scope of Nature Communications and can be considered for publication after addressing or clarification of the comments. My specific comments are below:

=> We thank the reviewer's encouraging comments. The reviewer evaluated that *"Overall the manuscript is in the scope of Nature Communications and can be considered for publication after addressing or clarification of the comments."*

=> We have now carefully revised the manuscript, and our response to the reviewer's helpful comments/suggestions are provided below. All main revisions are marked in red font.

1) line 112: It's a bit unclear why shift of crossover point to a lower frequency region means that coordinating anions render the crosslinking network more elastic and robust. Does it mean that the crosslinking network is more elastic when measuring properties at the same rate?

=> The rheological frequency sweep test provides rich information regarding to the viscoelastic properties of the material. From the experimentally determined storage (G') and loss (G'') moduli at variable frequencies, we gain insights into the time-dependent deformation behavior of the material. The storage modulus (G') reflects the amount of elastically stored energy in the material, thus indicating the elastic or solid-like portion. The loss modulus (G'') represents the energy dissipated as heat, indicating the viscous or liquid-like portion of the material. At a given frequency, the material exhibits 1) solid-like behavior when G' is greater than G'' ($G' > G''$), 2) liquid-like behavior when G' is lower G'' ($G' < G''$). The frequency at which G' and G'' have equal values (or at which $\tan(\delta) = G''/G' = 1$) is known as the crossover point or crossover frequency. At this crossover point, the material undergoes the transitions from solid-like to liquid-like behavior, or vice versa.

=> The shift of the crossover point to lower frequency or longer time region indicates that the material requires more time to start flow as a whole. This suggests that the material exhibits a high degree of resistance to deformation, indicating intrinsically more elastic and solid-like properties. In our study, we observed that the crossover point moves to a lower frequency as the coordinating ability of the counter anion increases. We concluded that the participation of

the coordinating anion in the ligand exchange process under the applied stress renders the entire polymer network elastic, which is in consistence with the results from our shear stress relaxation analysis and tensile tests.

=> The conventional frequency range for viscoelastic polymers is from 0.01 Hz to 100 Hz. For our polymers, we couldn't observe the crossover point within this frequency range, as the polymers are elastomers with predominantly elastic behavior. Therefore, we performed frequency sweep tests at different temperatures and employed time temperature superposition (TTS) principle to generate master curves. The frequency sweep at higher temperature allowed us to gain information at the lower frequency region which is practically challenging to access under ambient conditions. The underlying principle in TTS is that high temperature facilitates the movement of polymer chains, i.e. reduces the time required for the same deformation. By applying an appropriate shift factor, we could superimpose the experimental data obtained at high temperature onto the low frequency region of the results obtained at the reference temperature (in our case, 20 °C). This allowed us for a direct comparison of the crossover frequencies and corresponding elasticities of our polymers.

2) In terms of wording, it seems like bridging anions are grouped with coordinating anions creating some confusion at some instances for example in line 112 and 115. Should it be "bridging anions and coordinating anions" instead of "coordinating anions" in line 112 and 115?

=> We thank the reviewer for this suggestion. Before addressing this matter, we would like to inform the reviewer that we replaced the term "bridging anion" to "multimodal anion" due to the absence of direct experimental evidence of bridging coordination. On page 4 of our main text, we defined the multimodal anion as a coordinating anion capable of multiple coordination modes. The multimodal anions (OAc^- and acac^-) also have coordinating ability to the metal center, and can be considered as part of the coordinating anion. We agree that the use of the

term “coordinating anion” in our manuscript for both generally accepted terminology and one of our anion categories has caused confusion. In the section that the reviewer mentioned, we intended to use the term “coordinating anion” in a general sense to include all the anions with coordinating abilities. To address this ambiguity, we have made following modifications. In our revised manuscript, we now specify that “coordinating anion” denotes only the coordinating anion with a single coordination mode.

Main text, Page 6

Non-coordinating anions (OTf⁻, TFSI⁻) make the polymer network softer and more stretchable, whereas anions with coordinating ability (Cl⁻, OAc⁻, and acac⁻) give higher modulus and fracture strength. In addition, the residual strain after cyclic mechanical test with the strain of 50% was smaller as the coordination ability of the anion is greater (**Figs. 3c** and **3d**), suggesting that the polymer networks become more elastic with a strong binding anion.

Main text, Page 7

This means that the crosslinking network becomes more elastic and robust as the coordinating ability of the anion increases.

The value of flow activation energy was significantly higher for anions with coordinating ability (Cl⁻, OAc⁻, and acac⁻), indicating that it is harder to make the polymer network flow when the anion binds more tightly.

In stark contrast, when the anions (Cl⁻, OAc⁻, and acac⁻) are able to coordinate with the metal ion, the value of stress relaxes gently due to slow energy dissipations.

3) line 138: Authors mention that Zn-acac- and Zn-OAc-BPy-PDMS exhibit better stretchability than Zn-Cl-BPy-PDMS. It is clear as the highest strain in Fig 3a is larger for the former. However, it is difficult to see why Zn-Cl-BPy-PDMS is said to have higher tensile

strength than others. Could the authors please clarify as the peak stress for Zn-Cl-BPy-PDMS is slightly higher than Zn-OAc-BPy-PDMS?

=> We deeply appreciate the reviewer's comment. Upon careful examination of the experimental data for the tensile test and the cyclic loading-unloading test of Zn-Cl-BPy-PDMS, we found a critical oversight in the measurement of the stress-strain curve. We realized that the stress level of the loading profile during the cyclic test performed at a rate of $200\% \text{ min}^{-1}$ (**Fig. 3c**) is lower than the stress level of the tensile test (**Fig. 3a**) conducted at a rate of $100\% \text{ min}^{-1}$ (**Fig. r11**). It is counter-intuitive that the stress level of the tensile test is lower at a higher strain rate. In addition, the Young's modulus value of Zn-Cl-BPy-PDMS appeared to be abnormally high. We have thoroughly reviewed all the details in our experiments and concluded that an error (presumably the measurement of the film thickness) occurred when measuring the stress-strain curve of Zn-Cl-BPy-PDMS.

Fig. r11 | Original stress-strain curves of Zn-Cl-BPy-PDMS at different loading rates.

Solid line; at a loading rate of $100\% \text{ min}^{-1}$, dashed line; at a loading rate of $200\% \text{ min}^{-1}$.

=> To address this issue, we prepared the polymer films of Zn-Cl-BPy-PDMS more strictly and conducted the tensile tests again. The measurements were repeated for five times to ensure the accuracy (**Fig. r12**). Upon reevaluation, we observed that the newly prepared Zn-Cl-BPy-PDMS film exhibited a fracture strain of 65%, a Young's modulus of 1.4 MPa, and a tensile strength of 0.53 MPa, which aligns with the results obtained from other measurements (**Fig. 3a and Supplementary Table 1**). The weaker coordination of Cl⁻ anion compared to OAc⁻ anion imparts higher viscoelasticity to the polymer film, resulting in slightly lower tensile strength and Young's modulus. We sincerely thank the reviewer for bringing this apparent mistake to our attention, as it led us to reevaluate and obtain accurate results.

Fig. r12 | Stress–strain curves of Zn-Cl-BPy-PDMS with a sample width of 3 mm, a thickness of 0.2–0.4 mm, and a length of 5 mm at a loading rates of 5 mm min⁻¹. The measurement was repeated multiple times.

Fig. 3a | Effects of counter anions on mechanical and dynamic properties of Zn-X-BPy-PDMS. Stress–strain curves of Zn-X-BPy-PDMS polymer films (acac⁻; red, OAc⁻; orange, Cl⁻; gray, OTf⁻; blue, TFSI⁻; green) with a sample width of 3 mm, a thickness of 0.2–0.4 mm, and a length of 5 mm at a loading rate of 5 mm min⁻¹.

Entry	Counter anion	Young's modulus ^a (MPa)	Tensile strength (MPa)	Fracture strain (%)	Mechanical Toughness (x 10 ⁶ J m ⁻³)
1	acac ⁻	2.9	1.27	109	1.02
2	OAc ⁻	1.7	0.73	96	0.45
3	Cl ⁻	1.4	0.53	65	0.22
4	OTf ⁻	1.3	0.30	356	0.97
5	TFSI ⁻	0.86	0.22	675	1.41

Supplementary Table 1. Mechanical properties of Zn-X-BPy-PDMS polymer films under a loading rate of 100% min⁻¹. ^a calculated from the initial slope of stress–strain curves (within 5%).

4) In Fig 3f and Fig 5d, it is a bit unclear how the second peak is identified as the second peak is not visually apparent. Could the authors clarify the existence of second peak?

=> We thank the reviewer for this suggestion. The same point was raised by the reviewer #2 in his/her comment #10 regarding the unclear visualization of the relaxation time spectra. We have revised the figures accordingly. Please see the revised figures (Fig. 3f–h, Fig. 4e, Fig. 5d–f, Supplementary Fig. 5, and Supplementary Fig. 16c) for the updated visualization. We have included the relaxation time spectra of each coordination mode as dashed lines in all the relaxation time spectra. We hope that these modifications provide a clearer understanding of our findings.

Fig. 3f-h | bottom, Relaxation time spectra, $H(\tau)$ of **f**, Zn-acac-BPy-PDMS, **g**, Zn-Cl-BPy-PDMS, and **h**, Zn-OTf-BPy-PDMS. The dashed lines in the plot indicate the spectra of each energy dissipation mode.

Fig. 4e | bottom, Relaxation time spectrum, $H(\tau)$ of Zn-acac-OTf-BPy-PDMS which exhibits $acac^-$ -like broad and multiple peaks. The dashed lines in the plot indicate the spectra of each energy dissipation mode.

Fig. 5d-f | bottom, Relaxation time spectra, $H(\tau)$ of **d**, Cu- $acac^-$ -BPy-PDMS, **e**, Cu- OTf^- -BPy-PDMS, and **f**, Cu- $acac^-+OTf^-$ -BPy-PDMS. The dashed lines in the plot indicate the spectra of each energy dissipation mode.

Supplementary Fig. 5. bottom, Relaxation time spectra, $H(\tau)$ of **a**, Zn- OAc^- -BPy-PDMS, **b**, Zn- $TFSI^-$ -BPy-PDMS. The dashed lines in the plot indicate the spectra of each energy

dissipation mode.

Supplementary Fig. 16. c, bottom, Relaxation time spectrum, $H(\tau)$ of Zn-acac-TFSI-BPy-PDMS which exhibits acac⁻-like broad and multiple peaks. The dashed lines in the plot indicate the spectra of each energy dissipation mode.

5) typo: line 433 'on a TA instrument'

=> Correction was made accordingly. We thank the reviewer for pointing out this obvious oversight.

REVIEWERS' COMMENTS

Reviewer #1 (Remarks to the Author):

The author have improved the manuscript according to the reviewers' comments and suggestions. This work is impressive and solid enough now. The manuscript can be accepted without further revision.

Reviewer #2 (Remarks to the Author):

The authors have sufficiently addressed my concerns. I support publication without further modification.

Reviewer #3 (Remarks to the Author):

I thank the authors for carefully addressing my comments, performing additional synthesis/measurements to address certain concerns and revising their manuscript accordingly. I believe the manuscript has been significantly improved and can be accepted for publication.